# TriForces: Augmenting Atomistic GNNs for Transferable Representations

Ali Ramlaoui [1,2]   Alexandre Duval [1]   Hannah Bull [1]   Victor Schmidt [1]
Hugues Talbot [2]   Fragkiskos D. Malliaros [2]   Joseph Musielewicz [1]

## Abstract

Machine learning interatomic potentials (MLIPs) achieve excellent accuracy when trained on large Density Functional Theory (DFT) data. To be useful in practice, they must often be adapted to target chemistries using small and expensive task-specific datasets. However, MLIPs transfer inconsistently across domains, with representations that often lose accessible composition and structure information. To address this, we present TriForces, a model-agnostic three-stream framework that separates composition and structure information, combined with self-supervised learning to preserve transferable representations. TriForces improves performance on MatBench and QM9 over baselines without needing DFT labels and enables efficient similar structure retrieval through its learned latent space. On OMat24, in a limited-data training regime, TriForces reduces energy MAE by 57% at 20K samples only and improves force MAE across sample sizes. We release pretrained TriForces variants across multiple MLIP architectures with code at https://github.com/Ramlaoui/triforces.

## 1. Introduction

Density functional theory (DFT) (Kohn & Sham, 1965) provides a computational model to obtain atomistic properties like energies and forces, but remains expensive, typically scaling as $\mathcal{O}(n^3)$ with the number of electrons (Burke, 2012). Fueled by large-scale DFT datasets (Jain et al., 2020; Schmidt et al., 2024; Ramlaoui et al., 2025; Chanussot et al., 2021), geometric Graph Neural Networks (GNNs) (Duval et al., 2023a) now achieve remarkable accuracy on property

prediction tasks. Such Machine Learning Interatomic Potentials (MLIPs) can drive simulations and materials discovery workflows (Batatia et al., 2025; Friederich et al., 2021; Zeni et al., 2025) through geometry optimization, molecular dynamics, and exploration of reaction pathways (Wander et al., 2025).

However, the diversity and complexity of chemical systems require frequent adaptation of models to new properties, systems and DFT functionals, often using small costly task-specific datasets. Retraining from scratch is computationally expensive and does not transfer learned representations, while fine-tuning involves many parameters and often leads to catastrophic forgetting (Huang et al., 2025). We find that MLIPs pre-trained on 100M structures can fail to fine-tune even on simple diagnostic tasks, such as identifying the crystal system or predicting the majority element in a structure's composition (App. C.1). Transfer performance also varies substantially across tasks and benchmarks (Niblett et al., 2025), and practical choices, such as which layers to fine-tune (Pinto, 2025), how much data to use, or which pretrained model to start from, can significantly affect outcomes (Focassio et al., 2024).

A second limitation points to the same underlying issue: current MLIPs learn representations optimized for prediction rather than for reuse. In other domains, self-supervised learning (SSL) has shown that representations can preserve semantic structure enabling nearest-neighbour retrieval in embedding space for exploratory analysis (Caron et al., 2021; Chen et al., 2020; He et al., 2020; 2022). Such retrieval also offers practical advantage in materials science for comparing candidates and exploring structure-property relationships (Hu et al., 2019; Stärk et al., 2022; Yang et al., 2022). By contrast, training to regress energies and forces encourages representations sufficient for those targets but not organized to preserve composition and geometry. As a result, embeddings support retrieval poorly, and local neighbourhoods can be unstable across models or domains (Li & Walsh, 2025). We attribute both transfer instability and weak reuse to supervised objectives and standard architectures that entangle composition and geometry.

We introduce TriForces, a framework that makes atomistic representations easier to transfer by splitting them into three

[1]Entalpic, Paris, France [2]Université Paris-Saclay, Centrale-Supélec, Inria, Gif-sur-Yvette, France. Correspondence to: Ali Ramlaoui <ali.ramlaoui@entalpic.ai>, Fragkiskos D. Malliaros <fragkiskos.malliaros@centralesupelec.fr>, Joseph Musielewicz <joseph.musielewicz@entalpic.ai>.

*Proceedings of the 43$^{rd}$ International Conference on Machine Learning*, Seoul, South Korea. PMLR 306, 2026. Copyright 2026 by the author(s).

components: composition, structure, and interaction (Fig. 1). Instead of adding auxiliary features to a single latent vector, the composition stream encodes chemistry without coordinates, the structure stream encodes geometry without element identity, and a base geometric GNN captures their coupling. This factorization enables self-supervised objectives to play complementary roles: denoising can now emphasize geometric stability, masking can encourage learning compositional patterns, and non-reconstruction objectives (e.g., LeJEPA (Balestriero & LeCun, 2025), Barlow Twins (Zbontar et al., 2021)) improve latent separability. Beyond making fine-tuning easier and more efficient, the resulting embeddings support decomposed similarity search by chemistry, structure, or jointly.

We demonstrate TriForces on base geometric GNN architectures spanning equivariant (MACE (Batatia et al., 2022), eSEN (Fu et al., 2025) and non-equivariant (Orb-v3 (Rhodes et al., 2025)) designs. Our contributions are:

1. A three-stream decomposition augmenting existing atomistic architectures with compositional and structural pathways, preserving both factors by design.
2. A self-supervised pretraining strategy that combines reconstruction-based objectives with latent-prediction learning via LeJEPA to structure the embedding space and improve downstream transfer.
3. We enable interpretable similarity retrieval by performing search in the compositional, structural, or joint embedding spaces, supporting materials comparison and exploratory analysis.
4. Extensive experimental validation across multiple architectures and benchmarks, demonstrating improved data efficiency, transfer performance, and representation quality.

Our results show that stream factorization provides the dominant gains in large supervised settings, while SSL is most valuable for low-data transfer, representation organization, and retrieval. We therefore view TriForces as an architectural framework whose representations are further improved by SSL, rather than as an SSL method alone. Code, training recipes, and pretrained TriForces checkpoints are available at https://github.com/Ramlaoui/triforces.

## 2. Related Work

**Geometric models for atomistic systems.** Geometric deep learning has become the dominant paradigm for molecular and materials modeling. Equivariant architectures such as SchNet (Schütt et al., 2017), DimeNet (Gasteiger et al., 2020), MACE (Batatia et al., 2022), and EquiformerV2 (Liao et al., 2024b) incorporate physical symmetries directly into their design, while non-equivariant approaches, including Orb (Neumann et al.,

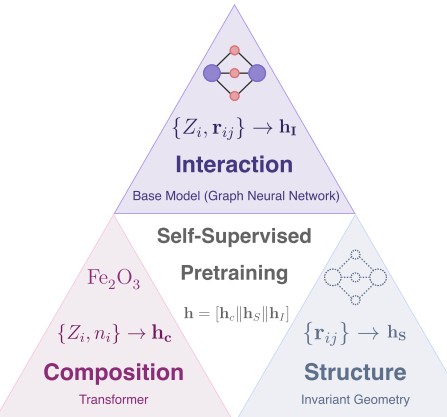

*Figure 1.* TriForces builds upon existing geometric GNNs by adding compositional and structural streams, as well as multi-objective self-supervised pretraining.

2024) and FAENet (Duval et al., 2023b), learn invariances from data through augmentation. These models achieve strong in-distribution accuracy and have enabled large-scale atomistic simulation and discovery workflows. However, architectural advances alone primarily improve predictive performance on the training task and do not explicitly encourage representations that are reusable across tasks and domains (Niblett et al., 2025; Pinto, 2025). Consequently, a growing literature explores fine-tuning strategies for better adaptation (Zhang et al., 2026; Kong et al., 2025), yet questions remain about reuse in analysis settings such as retrieval and similarity search.

**SSL for molecules and materials.** SSL has been explored for atomistic systems using a variety of objectives, including denoising (Zaidi et al., 2022; Neumann et al., 2024), masked reconstruction (Dong et al., 2025), and representation-level learning (Magar et al., 2022; Koker et al., 2022; Zhou et al., 2023). Denoising has also been used as an auxiliary objective alongside supervised training or as equilibrium-structure pretraining, notably in Noisy Nodes and extensions to non-equilibrium structures via force-aware denoising (Godwin et al., 2022; Liao et al., 2024a). Prior work has shown that SSL can improve data efficiency and fine-tuning performance on downstream tasks (Zaidi et al., 2022). However, these methods are typically evaluated in isolation and primarily through predictive accuracy, leaving open questions about how different SSL objective families affect representation organization and reuse. In particular, little attention has been paid to how SSL interacts with architectural inductive biases or whether learned representations support tasks beyond prediction, such as retrieval or exploratory analysis.

**Compositional representations.** A parallel line of work models materials using composition alone, without explicit

structural information. Models such as Roost (Goodall & Lee, 2020) and CrabNet (Wang et al., 2021) demonstrate that stoichiometry encodes strong global signals and can achieve competitive performance across a range of materials property prediction tasks. These results highlight the richness of compositional information, but also underscore its limitations: composition alone cannot capture geometric effects or distinguish polymorphs.

Building upon existing geometric models with strong performance on predictive tasks, the TriForces architecture includes explicit encoders for compositional and structural representations. Self-supervised training integrates denoising, masking, and representation-level objectives that target complementary aspects of the learned representations. This design is intended to encourage representations that are transferable across tasks and suitable for analysis settings such as probing and retrieval, in addition to downstream prediction.

## 3. TriForces Framework

TriForces, illustrated in Fig. 2, enhances existing geometric GNNs with (1) a three-stream architecture ensuring retention of compositional and structural information (Sec. 3.1), and (2) SSL representations that are useful for more accurate predictive models as well as for better generalization and interpretation (Sec. 3.2). We use **TriForces-Streams** for the three-stream architecture trained from random initialization, **TriForces** for its SSL-initialized version, and **Base + SSL** for backbone-only SSL. This separates stream factorization from SSL pretraining.

### 3.1. Three-Stream Architecture

Given an input structure $\mathcal{X} = (\{z_i\}, \{\mathbf{x}_i\})$ with atomic species $z_i$ and positions $\mathbf{x}_i$, represented by a graph $\mathcal{G}$ where nodes correspond to atoms and edges are constructed using a radius cutoff, TriForces produces the learnable node-level representation:

$$\mathbf{h}_i = \left[\mathbf{h}_i^{\text{comp}}, \ \mathbf{h}_i^{\text{struct}}, \ \mathbf{h}_i^{\text{int}}\right] \tag{1}$$

via three parallel streams detailed below.

**Compositional stream (no geometry).** The compositional stream encodes chemistry without coordinates. For each structure, the atoms list is compressed into a set of *unique elements* $\{(z_t, c_t)\}_{t=1}^{T}$, where $z_t$ is an atomic number and $c_t \in \mathbb{N}$ is its count in the structure ($T \ll N$ in practice). Tokens are initialized with a learnable element embedding,

$$\mathbf{u}_t = \mathbf{e}(z_t). \tag{2}$$

We then apply a Transformer over the $T$ tokens and inject stoichiometric information via *count-weighted attention*. For attention head $h$, queries and keys are defined as $\mathbf{q}_t^{(h)} =$

$\mathbf{W}_Q^{(h)}\mathbf{u}_t$, $\mathbf{k}_s^{(h)} = \mathbf{W}_K^{(h)}\mathbf{u}_s$. We modify the attention logits by adding a log-count bias:

$$a_{ts}^{(h)} = \frac{\left(\mathbf{q}_t^{(h)}\right)^\top \mathbf{k}_s^{(h)}}{\sqrt{d_h}} + \log(c_s),$$
$$\alpha_{ts}^{(h)} = \text{softmax}_s\left(a_{ts}^{(h)}\right). \tag{3}$$

This makes attention over unique element tokens equivalent to attention over all atoms of each type, while keeping computation $\mathcal{O}(T^2)$ and independent of $N$ as shown in Proposition A.1.

Unlike composition-only predictors that normalize stoichiometry to fractions or reduced formulas (Wang et al., 2021; Goodall & Lee, 2020), we preserve *absolute* element counts $c_t$, since they encode real physical information about system size and correlate, for example, with energy scale.

**Structural stream (type-agnostic geometry).** Many materials properties depend on recurring geometric motifs that are shared across different chemistries. By learning an atomic type-agnostic geometric representation, the structural stream captures reusable structural patterns and provides an inductive bias that complements the chemistry-aware interaction stream. Because it is not used to model directional responses, we restrict this stream to rotation-invariant features.

Given neighbour displacements $\mathbf{r}_{ij} = \mathbf{x}_j - \mathbf{x}_i$, distances $r_{ij}$, and directions $\hat{\mathbf{r}}_{ij}$, we construct a learnable, rotation-invariant local descriptor inspired by SOAP-style density representations (Bartók et al., 2013). Specifically, we expand each neighbour contribution using a radial basis $\{\phi_k(r)\}$ (Bessel or Gaussian RBF) (Schütt et al., 2017; Gasteiger et al., 2020), real spherical harmonics $\{Y_{lm}(\hat{\mathbf{r}})\}$ up to $l_{\max}$, and a set of smooth cutoff functions $\{s_s(r)\}_{s=1}^{S}$ for multi-scale geometry.

We add learnable parameters through a mixing of the radial and cutoff channels,

$$\tilde{\phi}_\alpha(r_{ij}) = \sum_{k,s} (\mathbf{W})_{\alpha,(k,s)} \, s_s(r_{ij}) \, \phi_k(r_{ij}), \tag{4}$$

which defines geometry-dependent density channels. Here $i$ indexes the central atom, $j \in \mathcal{N}(i)$ its neighbours, $k$ radial basis functions, $s$ cutoff scales, $l, m$ spherical harmonic indices, and $\alpha$ mixed radial-cutoff channels. For each atom $i$, we accumulate *density coefficients*

$$c_{\alpha lm}(i) = \sum_{j \in \mathcal{N}(i)} \tilde{\phi}_\alpha(r_{ij}) \, Y_{lm}(\hat{\mathbf{r}}_{ij}), \tag{5}$$

which correspond to a local neighbour density expanded in a spherical basis. Rotation invariance is enforced by forming the power spectrum:

$$p_{\alpha\alpha'l}(i) = \sum_m c_{\alpha lm}(i) \, c_{\alpha'lm}(i), \tag{6}$$

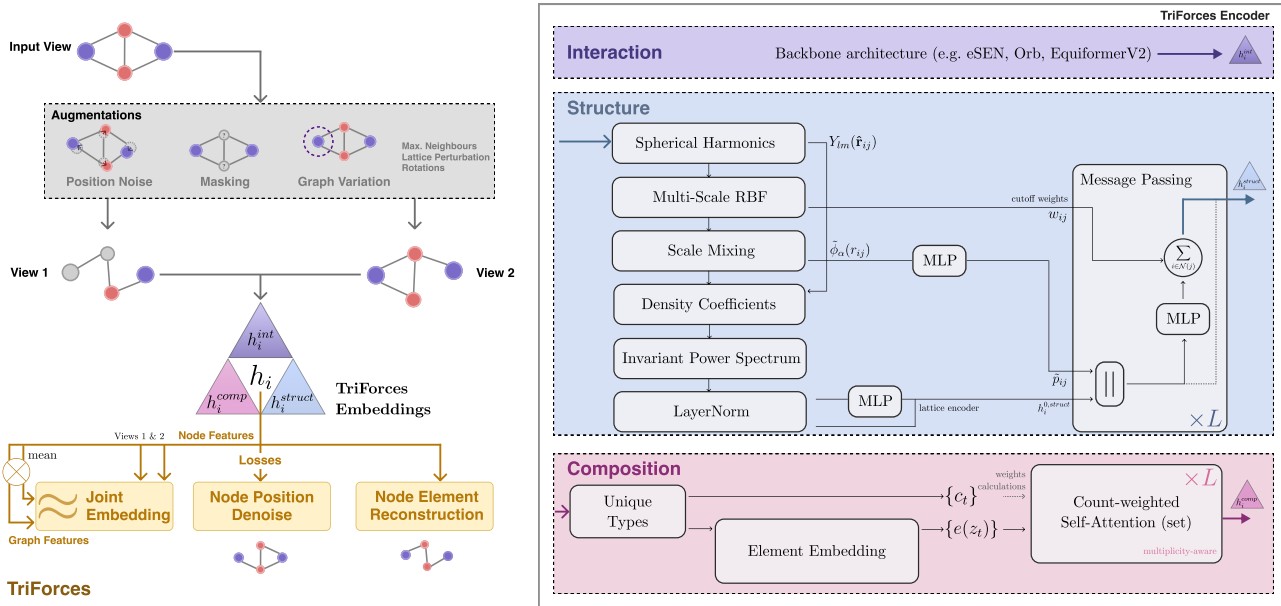

*Figure 2.* TriForces self-supervised pretraining and encoder components. Left: view 1 and view 2 are two independently sampled stochastic augmentations of the same input structure, constructed using position noise, element masking, graph variation, and rotations for non-equivariant backbones. These views support denoising, masked element reconstruction, and non-reconstruction losses acting on node and graph level embeddings. Right: interaction stream from a base geometric GNN, structure stream via invariant power-spectrum features and message passing and a composition transformer with count-weighted attention. The three embeddings are concatenated to form the TriForces features.

which is invariant to global rotations and translations by construction and equivalent to the standard SOAP power spectrum (Bartók et al., 2013). The resulting invariant features are flattened, and mapped through a small MLP with a residual connection to obtain, for each node, the initial representation $\mathbf{h}_i^{\text{struct}}$. For periodic crystals, we concatenate a learned lattice embedding (on normalized cell lengths and angles) broadcast to atoms to provide global periodicity context in addition to the radius cutoff. We then apply a small number of *invariant message passing* layers to incorporate connectivity and topology: each node aggregates messages from neighbours as a learned function of the neighbor state and rotation-invariant edge features, as detailed in App. A.2.

**Interaction stream.** The interaction stream captures how composition and geometry jointly determine properties, preserving the expressiveness of the base model. It is typically a geometric GNN (Duval et al., 2023a) with full access to both species as node features and positions:

$$\{\mathbf{h}_i^{\text{int}}\} = \text{GNN}_{\text{base}}\big(\{z_i\}, \{\mathbf{x}_i\}\big). \tag{7}$$

This can be any existing architecture that operates on arrangements of atoms. We demonstrate the improvements with Orb-v3, eSEN, and MACE.

**Node features.** The three streams are concatenated to form the final node representation,

$$\mathbf{h}_i = [\mathbf{h}_i^{\text{comp}} \parallel \mathbf{h}_i^{\text{struct}} \parallel \mathbf{h}_i^{\text{int}}].$$

This modular design benefits transfer: tasks depending primarily on composition or geometry can leverage the corresponding streams directly, while tasks requiring both use the full representation.

### 3.2. Self-Supervised Pretraining

Architectural separation alone does not guarantee well-organized or transferable representations. We therefore employ self-supervised pretraining to shape the representation space before task-specific supervision, with the goals of stabilizing geometric information, reinforcing chemical correlations, and improving latent representation. We combine three complementary objectives: a non-reconstruction objective that aligns representations across augmented views, a denoising objective that stabilizes geometric representations, and a masking objective that encourages learning compositional context.

**Augmentations.** All self-supervised objectives operate on the same set of stochastic augmentations. For each input structure, we sample two augmented views by applying a shared augmentation pipeline that includes atomic position noise, atom-type masking, and randomized graph construction parameters. This design ensures that all objectives act on compatible perturbations and jointly shape the learned representations. We denote $\tilde{\mathcal{G}}$, the corrupted graph obtained after stochastically applying the augmen-

tations. We denote the two corrupted views as $\tilde{\mathcal{G}}_1$ and $\tilde{\mathcal{G}}_2$. Each view applies atom-type masking, Gaussian position perturbations, randomized graph construction through cut-off and max-neighbor changes, and random rotations for non-equivariant backbones. Since these augmentations preserve node identity, node-level embeddings can be aligned one-to-one across views.

**Non-reconstruction objective.** Non-reconstruction objectives have proven effective in self-supervised representation learning by encouraging invariance to data augmentations while avoiding representation collapse (Chen et al., 2020; He et al., 2020; Grill et al., 2020). Rather than reconstructing inputs, the non-reconstruction objective aligns embeddings from multiple augmented views of the same structure using LeJEPA (Balestriero & LeCun, 2025).

Since augmentations do not add or remove nodes, we can apply the loss at both the node level (one-to-one correspondence) and the graph level (mean-pooled features):

$$\mathcal{L}_{\text{LeJEPA}} = \mathcal{L}^{\text{g}} + \mathcal{L}^{\text{n}}, \tag{8}$$

where each term combines a SIGReg (Balestriero & Le-Cun, 2025) regularizer that enforces an isotropic Gaussian distribution via random projections, eliminating the need for stop-gradient or momentum encoders while preventing collapse:

$$\mathcal{L}^{\text{g}} = \lambda \mathcal{L}_{\text{SIG}}(\bar{\mathbf{h}}) + (1 - \lambda)\|\bar{\mathbf{h}}_1 - \bar{\mathbf{h}}_2\|^2 \tag{9}$$

$$\mathcal{L}^{\text{n}} = \lambda \mathcal{L}_{\text{SIG}}(\mathbf{h}) + (1 - \lambda)\sum_i \|\mathbf{h}_{1,i} - \mathbf{h}_{2,i}\|^2. \tag{10}$$

In Eq. 9, subscripts 1 and 2 denote embeddings computed from the two augmented views $\tilde{\mathcal{G}}_1$ and $\tilde{\mathcal{G}}_2$, respectively. Here, $\mathcal{L}^{\text{g}}$ and $\mathcal{L}^{\text{n}}$ are graph-level and node-level losses, $\bar{\mathbf{h}}$ denotes mean-pooled features, $\mathbf{h}_i$ individual node embeddings, and $\lambda$ the regularization weight. While this objective promotes invariance across augmentations and improves global organization of the latent space, it does not explicitly preserve fine-grained geometric or compositional distinctions. We therefore complement it with reconstruction-based objectives: denoising and masking.

**Denoising.** As shown in Zaidi et al. (2022), training the model to recover clean atomic configurations from perturbed inputs can be seen as a closely related problem to training a force field. We corrupt atomic positions by adding Gaussian noise: $\tilde{\mathbf{x}}_i = \mathbf{x}_i + \boldsymbol{\epsilon}_i, \boldsymbol{\epsilon}_i \sim \mathcal{N}(0, \sigma^2\mathbf{I})$, then construct a graph based on those new positions $\tilde{\mathcal{G}}$. The model predicts the noise (or equivalently, the clean positions):

$$\mathcal{L}_{\text{denoise}} = \sum_i \|f_\theta(\tilde{\mathcal{G}})_i - \boldsymbol{\epsilon}_i\|^2, \tag{11}$$

where $f_\theta(\tilde{\mathcal{G}})_i$ is the predicted noise for node $i$. Denoising encourages learning stable geometric configurations. For non-equivariant models, it also provides implicit rotation

augmentation: the model sees structures in diverse orientations and must learn to predict consistent outputs, aligning its latent space with orientation-related information. For simplicity, we employ a Mean Squared Error (MSE) objective rather than a diffusion loss function (Neumann et al., 2024), which we found to be sufficient in practice.

**Masking.** Masked atom prediction forces the model to infer element identity from surrounding atoms and geometry, rather than memorizing isolated embeddings. Following Dong et al. (2025), we find that predicting the atomic numbers of masked nodes improves downstream linear probing performance on properties like energy per atom. This task also benefits the composition stream by helping it learn patterns of elements that frequently co-occur. We randomly mask a fraction $p \sim \mathcal{U}[p_{\min}, p_{\max}]$ of nodes by replacing their atom embeddings with a learnable mask token. A classification head $g_\theta$ predicts the original atom types:

$$\mathcal{L}_{\text{mask}} = -\sum_{i \in \mathcal{M}} \log p_\theta(z_i \mid \tilde{\mathcal{G}}), \tag{12}$$

where $\mathcal{M}$ is the set of masked nodes, $z_i$ is the original atomic number and $\tilde{\mathcal{G}}$ is the corrupted graph where masking and other augmentations have been applied.

**Combined objective.** Our final pretraining loss combines all three terms:

$$\mathcal{L} = \mathcal{L}_{\text{denoise}} + \lambda_{\text{mask}}\mathcal{L}_{\text{mask}} + \lambda_{\text{LeJEPA}}\mathcal{L}_{\text{LeJEPA}}. \tag{13}$$

As we show in Sec. 4.6, the objectives are complementary and combining them with the additional streams gives further downstream gains, highlighting their interaction.

# 4. Experiments and Results

We pre-train TriForces on the LeMat-Bulk (Siron et al., 2025) dataset, containing 5M bulk structures with a diverse coverage of the periodic table starting with randomly initialized weights of eSEN, Orb-v3 and MACE. Depending on the experiment, we additionally fine-tune TriForces for downstream tasks. The remainder of this section presents experiments and results of transfer learning with TriForces of OMat24 (Sec. 4.1), MatBench (Dunn et al., 2020) (Sec. 4.2), and QM9 (Ramakrishnan et al., 2014) (Sec. 4.3). We additionally report linear probing metrics (Sec. 4.4), examples of similarity retrieval (Sec. 4.5), and ablation studies (Sec. 4.6).

## 4.1. Comparison to MLIP Benchmarks

**Fine-tuning.** We first test whether TriForces improves supervised MLIP fine-tuning for energy, force, and stress prediction. Table 1 summarizes fine-tuned performance on a 4M subset of OMat24 (Barroso-Luque et al., 2024) for each base architecture, separating the effect of adding

streams from the additional effect of SSL pretraining. Tri-Forces improves transfer across all architectures on energy, forces, and stress. For example, on Orb-v3 conservative, TriForces reduces OMat24 energy per atom MAE from 107 to 19.4 meV/atom and force MAE from 150 to 95.5 meV/Å without sacrificing force MAE. We find the largest gains on energy-conserving models (i.e., where forces are obtained as the gradients of the energy with respect to the positions). Fine-tuning details lie in App. B.1.

**Energy and Forces coupling.** We hypothesize these gains to happen because energy and force losses provide coupled gradients during conservative training, which can compete during optimization. Prior works mitigate this with staged schedules or specialized initialization. For e.g., MACE (Batatia et al., 2022) uses a two-step energy-weighting procedure where forces are trained initially and then the energy-loss weight is increased in a second stage, eSEN (Fu et al., 2025) trains for half of the steps with direct force training before enabling energy-conservation, and Orb (Neumann et al., 2024) uses diffusion pretraining. The three-stream TriForces architecture reduces the need for such tricks: we can train conservative models directly while reaching low energy MAE quickly without sacrificing force MAE or having to tune loss coefficients. The composition stream adds force-preserving degrees of freedom without introducing new position dependence. Appendix A.4 provides a theoretical intuition using rank-based bound connecting to this observation.

**Data efficiency.** Fig. 3 reports data efficiency on OMat24 fine-tuning with a fixed training budget across dataset sizes. TriForces improves energy and force MAE at every sample size, with the largest gains in the low-data regime: at 20K samples, energy per atom MAE drops from 81.3 to 34.6 meV/atom. Full results are provided in App. B.2.

**Large supervised scale.** At full OMat24 scale, SSL is no longer the main source of final accuracy gains. In a preliminary full-scale run, SSL only marginally improves final error but improves convergence speed, suggesting that SSL is most useful in transfer and low-data regimes. App. B.3 further compares SSL and supervised pretraining on matched data.

**MatBench Discovery.** MatBench Discovery (Riebesell et al., 2023) evaluates a model's ability to identify stable materials by predicting relaxed energies and classifying distance-to-hull against the Materials Project (Jain et al., 2020) structures. Following standard practices in Barroso-Luque et al. (2024), we train TriForces-pretrained variants of eSEN-sm and Orb-v3 on OMat24, then MPtrj (Deng et al., 2023) and sAlex; fine-tuning details are in Table 21. Results on the unique-prototypes test set are shown in Table 3. Tri-Forces Orb-v3 improves MAE (0.024 to 0.020 eV/atom) and $R^2$ (0.821 to 0.874) with slight gains in F1/Acc. TriForces

*Table 1.* Fine-tuning on OMat24 (4M subset) for 2 epochs across architectures and training modes (conservative vs direct). We report MAE ($\downarrow$) on the OMat24 validation set and an MPtrj subset for energy per atom $E$, forces $F$, and stress $\sigma$ (units: meV/atom, meV/Å, meV/Å$^3$). **Bold**: best within each architecture. Adding TriForces-Streams/SSL improves most metrics across all architectures, with the full TriForces variant significantly better for conservative models.

| Model | OMat24 | | | MPtrj | | |
|---|---|---|---|---|---|---|
| | $E\downarrow$ | $F\downarrow$ | $\sigma\downarrow$ | $E\downarrow$ | $F\downarrow$ | $\sigma\downarrow$ |
| *Orb-v3 Conservative* | | | | | | |
| Baseline | 107 | 150 | 7.8 | 114 | 125 | 11.0 |
| + TriForces-Streams | 35.6 | 149 | 6.2 | 94.9 | 125 | 10.9 |
| + TriForces | **19.4** | **95.5** | **4.7** | **66.8** | **98.4** | **10.2** |
| *Orb-v3 Direct* | | | | | | |
| Baseline | 39.4 | 108 | 8.0 | 87.6 | 85.2 | 10.5 |
| + TriForces-Streams | 23.9 | 114 | **6.5** | 63.3 | **76.7** | **9.9** |
| + TriForces | **21.8** | **102** | 7.8 | **59.0** | 85.9 | 10.9 |
| *eSEN (equivariant)* | | | | | | |
| Baseline | 104 | 80.3 | 6.3 | 105 | **75.9** | 10.1 |
| + TriForces-Streams | 20.8 | 84.2 | 4.5 | 62.8 | 93.3 | 10.2 |
| + TriForces | **18.8** | **78.0** | **4.4** | **60.3** | 85.3 | **10.1** |
| *eSEN Direct (equivariant)* | | | | | | |
| Baseline | 48.2 | 94.6 | 11.7 | 97.1 | 101 | **8.6** |
| + TriForces-Streams | 20.0 | 96.3 | 11.7 | **62.2** | 103 | 8.7 |
| + TriForces | **18.8** | **90.2** | **11.6** | 63.0 | **86.9** | 8.7 |
| *MACE (equivariant)* | | | | | | |
| Baseline | 117 | 150 | 8.1 | 125 | 189 | 12.6 |
| + TriForces-Streams | 81.6 | 149 | 8.5 | 120 | 186 | 12.8 |
| + TriForces | **34.3** | **142** | **6.1** | **96.1** | **176** | **12.3** |

*Table 2.* Full OMat24 supervised fine-tuning with and without SSL initialization. At this scale, SSL has marginal final-error impact, while the three-stream architecture remains competitive.

| Model (backbone) | $E\downarrow$ meV/atom | $F\downarrow$ meV/Å |
|---|---|---|
| TriForces-Streams (eSEN) | 13.7 | 62.2 |
| TriForces (eSEN) | 13.5 | 62.1 |

eSEN-sm almost matches the larger eSEN-30M-OAM with 60% fewer parameters. Full metrics are in Table 11.

### 4.2. Comparison to MatBench Benchmarks

We report MatBench (Dunn et al., 2020) results in Table 4, using the standard splits mean over the 5-folds MAE (regression) and F1 (classification). We compare against both self-supervised baselines and DFT-labeled pretraining ($\dagger$); the DFT variants are the fine-tuned base models checkpoints discussed in Sec. 4.1. TriForces variants achieve the best overall result on 6/8 tasks; for example, Phonons MAE improves from 57.8 to 19.5 cm$^{-1}$ with TriForces eSEN.

TriForces variants are consistently better than baselines across tasks, with multiple best or near-best results while being fully self-supervised. DFT-pretrained models remain

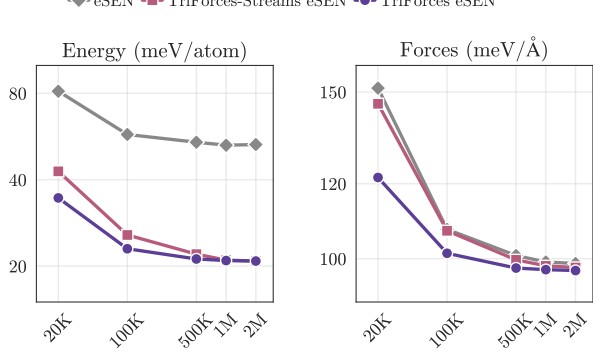

*Figure 3.* Data efficiency on OMat24 fine-tuning with a fixed training budget across dataset sizes (20K–2M samples). Curves show energy per atom and force MAE ($\downarrow$). TriForces-Streams uses random initialization, while TriForces uses SSL initialization. TriForces improves energy and forces at every data size, with the largest gains in the low-data regime.

*Table 3.* Matbench Discovery benchmark results (unique prototypes test set) comparing baselines with TriForces variants. F1 score; MAE (eV/atom); RMSD: root mean square displacement. TriForces eSEN achieves comparable energy MAE to the larger eSEN variant while training up to $\times 5$ faster.

| Model | F1 | Acc | MAE | $R^2$ | RMSD | Params |
|---|---|---|---|---|---|---|
| eSEN-30M-OAM | **0.925** | **0.977** | **0.018** | 0.866 | **0.061** | 30.2M |
| TriForces eSEN-sm | 0.915 | 0.974 | 0.019 | **0.874** | 0.062 | 12M |
| Orb-v3 | 0.905 | 0.971 | 0.024 | 0.821 | 0.075 | 25.5M |
| TriForces Orb-v3 | 0.910 | 0.972 | 0.020 | 0.874 | 0.065 | 42M |

best on some targets, but TriForces narrows the gap, showing the promises of stream decomposition plus SSL.

To separate stream factorization from parameter count, we also compare against parameter-matched widened backbones in App. B.7. TriForces remains better on all 8 MatBench tasks for both eSEN and Orb and on 6/7 QM9 targets for eSEN, indicating that the gains are not explained by capacity alone.

### 4.3. Comparison to QM9 Benchmarks

We evaluate QM9 molecular property prediction (Ramakrishnan et al., 2014; Ruddigkeit et al., 2012) for eSEN under different SSL pretraining datasets; results for a representative subset of targets are in Table 5. With OMol25 SSL pretraining, TriForces improves $\mu$ MAE from 0.023 to 0.018 D and $\alpha$ MAE from 0.098 to 0.075 $a_0^3$. Full results in App. B.5.

Pretraining on molecule-containing data (OMol25 (Levine et al., 2025) or adding bulk materials) consistently improves over bulk-only and no-SSL baselines. This hints that combining multiple modalities in the pretraining dataset does not have negative effects and we can consider fully pre-training on diverse chemical inputs.

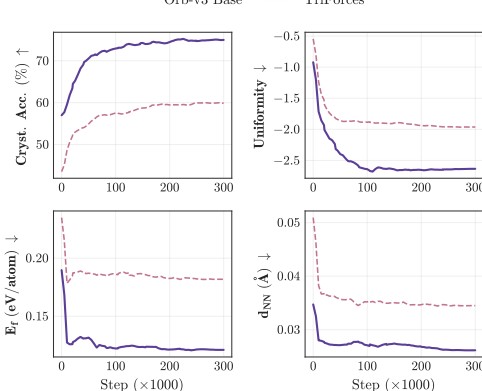

*Figure 4.* Linear probing metrics during SSL pretraining for Orb-v3 base vs TriForces variants (frozen embeddings). Both curves use the same SSL objectives; Orb-v3 Base is the backbone alone, while TriForces augments the backbone with composition and structure streams. We track formation-energy MAE, crystal-system accuracy, mean nearest-neighbour distance MAE, and uniformity. TriForces improves both compositional/structural probes and latent-space organization compared to the single-stream baseline.

### 4.4. Linear Probing

A common evaluation of self-supervised training in fields like vision involves using frozen embeddings of the trained model and measuring linear or non-linear regression on evaluation datasets, to assess the usefulness of the latent representations. While this has been less explored for materials embeddings, we track probing metrics during pretraining in Fig. 4. This comparison isolates architecture under identical SSL training: Orb-v3 Base is the backbone alone, while TriForces uses the same backbone with the two additional streams. Under the same SSL training framework, TriForces variants that add composition and structure streams improve crystal system accuracy while reducing formation-energy error, nearest-neighbour distance, and uniformity (Wang & Isola, 2020) compared to the base model.

### 4.5. Interpretable Similarity Retrieval

The three-stream architecture enables targeted similarity search. Given a query structure, we retrieve similar materials using cosine similarity on different stream embeddings: compositional similarity $\text{sim}(\mathbf{h}_q^{\text{comp}}, \mathbf{h}_d^{\text{comp}})$ finds materials with similar chemistry regardless of structure, structural similarity $\text{sim}(\mathbf{h}_q^{\text{struct}}, \mathbf{h}_d^{\text{struct}})$ finds geometrically similar materials regardless of composition, and joint similarity uses the full embedding. This decomposition is not possible in standard single-stream models where composition and structure are entangled.

Figure 5 evaluates $k$-NN retrieval using element set and space group as discrete targets, across different TriForces embedding streams. The composition stream performs best for element-set recall, while the structure stream performs

*Table 4.* MatBench results comparing TriForces pre-training against baselines. TriForces denotes the full method with self-supervised pretraining on top of the three-stream architecture. MAE (↓) / F1 (↑), mean over the 5 folds. **Bold**: best within category; underline: best overall. [†]: pre-trained with DFT labels.

| Task (Units) | JMP-L[†] | coGN | MACE[†] | TriForces MACE | Orb[†] | TriForces Orb | eSEN[†] | TriForces eSEN |
|---|---|---|---|---|---|---|---|---|
| Phonons ($cm^{-1}$) | **20.6** | 29.7 | 36.7 | **27.6** | 26.2 | **22.6** | 57.8 | **19.5** |
| Dielectric | **0.249** | 0.309 | 0.279 | **0.159** | 0.251 | **0.230** | **0.205** | 0.232 |
| Log GVRH ($\log_{10}$(GPa)) | **0.059** | 0.069 | 0.082 | 0.073 | 0.063 | **0.058** | 0.093 | **0.058** |
| Log KVRH ($\log_{10}$(GPa)) | **0.045** | 0.054 | 0.055 | 0.050 | 0.051 | **0.045** | 0.072 | **0.043** |
| Perovskites (meV) | **26.0** | 27.0 | 61.4 | **35.1** | 30.7 | **26.5** | 40.1 | **25.6** |
| MP Gap (eV) | **0.091** | 0.156 | 0.370 | **0.250** | 0.194 | **0.132** | 0.392 | 0.139 |
| MP E Form (meV/atom) | **10.1** | 17.0 | 40.8 | **34.4** | 21.1 | **17.1** | 83.5 | 20.2 |
| MP Is Metal (F1) | – | **0.901** | 0.858 | **0.876** | 0.895 | **0.902** | 0.811 | **0.888** |

*Table 5.* QM9 molecular property prediction using the split in Anderson et al. (2019). Models differ by SSL pretraining data: none (eSEN), crystalline-bulks only, LeMat-Bulk (Siron et al., 2025) + OMol25 (Levine et al., 2025), or molecules only. MAE (↓). **Bold**: best. Pretraining on molecule-containing data (OMol25 or Both) generally lowers MAE vs bulk-only or no SSL.

| Model | $\mu$ (D) | $\alpha$ ($a_0^3$) | $\varepsilon_{HOMO}$ (meV) | $\varepsilon_{LUMO}$ (meV) | $C_v$ (cal/mol K) | $U_0$ (meV) |
|---|---|---|---|---|---|---|
| eSEN | 0.023 | 0.098 | 23.6 | 25.1 | 0.037 | 9.0 |
| TriForces eSEN (Bulk) | 0.021 | 0.093 | 23.7 | 25.2 | 0.041 | 9.3 |
| TriForces eSEN (All) | 0.018 | 0.079 | 21.0 | 21.0 | 0.032 | **7.6** |
| TriForces eSEN (OMol) | **0.018** | **0.075** | **20.2** | **20.4** | **0.031** | 8.9 |

best for space-group recall, matching their intended semantics. Interaction streams track the baseline models. Further qualitative examples are in App. C.

### 4.6. Further Ablations

Table 6 summarizes a sequential stream ablation on OMat24-2M without SSL, using an energy-conserving eSEN-sm backbone. Full SSL-objective and stream ablations are provided in App. B.6.

**Impact of three streams.** The streams are complementary rather than uniformly redundant: composition primarily improves energy prediction, interaction is critical for forces and stress, and structure contributes most to geometric probing and representation quality. Removing the compositional stream most strongly degrades formation energy and fine-tuning energy/forces, while removing the structural stream most hurts crystal-system probing and uniformity, highlighting the complementarity of the two streams. Removing both streams collapses fine-tuning energy.

**Impact of SSL objectives.** LeJEPA-only fine-tuning underperforms while giving good uniformity metrics and probing results on geometric properties, indicating that stream separation, denoising, masking, and LeJEPA are complementary: denoising stabilizes geometry, masking improves compositional probes, and LeJEPA organizes the latent space for retrieval and transfer. Alternative SSL objectives (swapping LeJEPA with Barlow Twins (Zbontar et al., 2021), Crystal Twins (Magar et al., 2022), or denoising-only pretraining (Zaidi et al., 2022)) show mixed behavior and do not

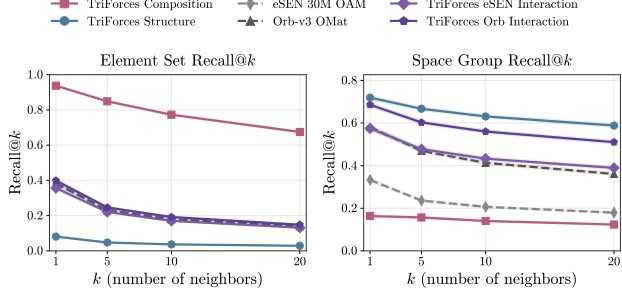

*Figure 5.* Nearest-neighbour recall@$k$ for element-set retrieval (left) and space-group retrieval (right) using different embedding streams. The composition stream is best for element-set recall, the structural stream is best for space-group recall, and the interaction stream tracks DFT-pretrained baselines.

match TriForces across the full set of metrics but improves forces prediction comparatively to the full TriForces.

## 5. Discussion

TriForces targets two pain points in atomistic foundation models: brittle transfer across tasks and representations that are hard to reuse (Niblett et al., 2025; Pinto, 2025; Focassio et al., 2024; Li & Walsh, 2025). The stream decomposition appears to do more than add capacity: gains are strongest when a stream aligns with the target signal, while the interaction stream often tracks the base model.

**Architecture versus SSL.** The three-stream architecture accounts for most gains in large-scale supervised settings, while SSL provides its largest benefits in low-data transfer, representation organization, and retrieval. This regime dependence is important practically: when abundant supervised labels are available, SSL is not essential for final predictive accuracy, but it still improves initialization and produces embeddings that are more useful for analysis.

**Pretraining data and non-equilibrium structures.** Both trajectory-rich, non-equilibrium structures and equilibrium structures can be used with TriForces (Siron et al., 2025; Liao et al., 2024a; Neumann et al., 2024), and matching the pretraining distribution to the target regime remains important (e.g., bulk pretraining may not transfer to molecules

*Table 6.* Sequential stream ablation on OMat24-2M without SSL using an energy-conserving eSEN-sm backbone. Each transition adds one stream. C, S, and I denote composition, structure, and interaction streams. Metrics are reported in meV/atom for $E$ and probe $E$, meV/Å for $F$, meV/Å$^3$ for $\sigma$, and percentage for Cry. Composition mainly improves energy, interaction is essential for forces and stress, and structure improves geometric probing.

| Transition | $E \downarrow$ | $F \downarrow$ | $\sigma \downarrow$ | Cry $\uparrow$ | Probe $E \downarrow$ |
|---|---|---|---|---|---|
| I $\rightarrow$ C+I | 102$\rightarrow$16 | 79$\rightarrow$77 | 6.1$\rightarrow$4.3 | 44.5$\rightarrow$50.6 | 61.4$\rightarrow$55.0 |
| C+S $\rightarrow$ Full | 68$\rightarrow$16 | 249$\rightarrow$78 | 8.9$\rightarrow$4.4 | 66.1$\rightarrow$68.1 | 126.6$\rightarrow$55.4 |
| C+I $\rightarrow$ Full | 16$\rightarrow$16 | 77$\rightarrow$76 | 4.3$\rightarrow$4.3 | 50.6$\rightarrow$68.1 | 55.0$\rightarrow$55.4 |

as the latter do not use periodic boundary conditions). The advantage is that pretraining data can be constructed without labels or costly computations and benefits from diversity in the input data.

**Interpretable retrieval and analysis.** The compositional and structural streams make the learned representation reusable beyond property prediction: they support chemistry-only retrieval without entangling geometry, and vice versa (Hu et al., 2019; Stärk et al., 2022). This enables targeted comparison of candidate materials in embedding space, while learning better fusion than concatenation without hurting the SSL objectives remains an open question.

**Computational considerations.** Extra streams add parameters and compute, with the structural stream dominating overhead. However, these added parameters are comparatively lightweight versus scaling the interaction backbone (they are coordinate-free or invariant), and parameter and compute-matched comparisons show the gains are not a simple parameter trade-off. App. D.2 summarizes these cost comparisons.

**Limitations and future work.** Our retrieval analysis is limited to bulk crystals and molecules; surfaces, defects, and complex interfaces may behave differently. Future work could explore these regimes, richer fusion between streams, and study scaling to larger models and datasets. Although this work trains TriForces as a pretraining framework, the additional streams could also be used on already-trained MLIPs by freezing or lightly adapting the interaction backbone, training the composition and structure streams, and then fine-tuning the combined model. Finally, the factorized representation is a natural candidate encoder for generative models such as diffusion or flow-matching models, where separating composition and geometry may improve controllability; we leave this direction to future work.

## 6. Conclusion

We introduced TriForces, a model-agnostic three-stream augmentation combined with multi-objective self-supervised pretraining that consistently improves accuracy, data efficiency, and interpretability across datasets and backbones (Orb-v3, eSEN, MACE). Its plug-in design makes it immediately applicable to new or modality-specific architectures while enabling chemistry and structure-aware retrieval. We released pretrained checkpoints and training recipes to support reuse and further study in downstream atomistic modeling.

## Acknowledgements

This project was provided with computer and storage resources by GENCI at CINES and IDRIS under the allocations 2025-AD011015800, 2025-A0181016212, 2025-AD011016353, and 2026-A0201016212 on the Jean Zay and Adastra supercomputers. This work was also performed using computational resources from the "Mésocentre" computing center of Université Paris-Saclay, Centrale-Supélec and École Normale Supérieure Paris-Saclay supported by CNRS and Région Île-de-France https://mesocentre.universite-paris-saclay.fr/.

## Impact Statement

This paper presents work whose goal is to advance machine learning methods for atomistic and materials modeling. Advances in atomistic and materials modelling can enable beneficial applications such as clean energy technologies, improved catalysts, and safer structural materials, though they may also contribute to the development of materials with harmful or dual-use applications. The methods presented in this work operate at a general modelling level and do not target any specific application domain or material class. We support responsible use of these methods.

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

## Summary of the Appendix

This appendix provides supplementary material organized as follows. Appendix A details the composition and structural stream architectures and presents theoretical analysis of energy and force gradient coupling. Appendix B reports additional experimental results including ablations, QM9 benchmarks, and data efficiency studies. Appendix C analyzes learned representations through probing, visualization, and retrieval examples. Appendix D lists hyperparameters, computational costs, pipeline summaries, and pseudocode for pretraining and fine-tuning.

## A. Architecture and Theory

### A.1. Composition Stream Details

We use the same composition stream across all base architectures unless otherwise noted. Table 7 summarizes shared hyperparameters. We denote the composition stream width $d_{\text{comp}}$, number of layers $L_{\text{comp}}$, attention heads $H_{\text{comp}}$, FFN width $d_{\text{ff}}$, dropout $p$, and element vocabulary size $|\mathcal{Z}|$. Proposition A.1 shows the equivalence between using the unique set of elements and the multiset.

**Proposition A.1** (Count-weighted attention equals attention over repeated tokens)**.** *Let* $\{(u_s, c_s)\}_{s=1}^T$ *be unique element tokens with counts* $c_s \in \mathbb{N}$. *Fix a query token* $t$ *and define logits* $\ell_{ts} \in \mathbb{R}$ *and values* $v_s$ *(for one attention head). Consider (i)* expanded *attention over a multiset where token* $s$ *is repeated* $c_s$ *times, each copy having the same logit* $\ell_{ts}$ *and value* $v_s$, *and (ii)* compressed *attention over unique tokens with log-count bias:*

$$\alpha_{ts} = \text{softmax}_s(\ell_{ts} + \log c_s).$$

*Then the attention outputs are identical:*

$$\sum_{j \in \text{expanded}} \text{softmax}_j(\ell_{t,\tau(j)}) \, v_{\tau(j)} \; = \; \sum_{s=1}^T \alpha_{ts} \, v_s,$$

*where* $\tau(j)$ *maps an expanded index to its token type.*

*Proof.* Each copy of token $s$ is identical because embeddings $\mathbf{u}_t = \mathbf{e}(z_t)$ only depend on $t$. In the expanded softmax, the total unnormalized mass assigned to type $s$ is $c_s e^{\ell_{ts}}$ (there are $c_s$ identical copies), and the normalizer is $\sum_{r=1}^T c_r e^{\ell_{tr}}$. Hence the total probability mass for type $s$ is

$$\forall\, s \in [1, T], \; \alpha_{ts}^{\text{cw}} = \frac{c_s e^{\ell_{ts}}}{\sum_r c_r e^{\ell_{tr}}} = \frac{e^{\ell_{ts} + \log c_s}}{\sum_r e^{\ell_{tr} + \log c_r}} = \text{softmax}_s(\ell_{ts} + \log c_s),$$

and multiplying by $v_s$ and summing over $s$ gives the same output as the expanded attention. $\square$

*Table 7.* Common composition stream hyperparameters used across TriForces models.

| Hyperparameter | Value |
|---|---|
| $d_{\text{comp}}$ (width) | 256 |
| $L_{\text{comp}}$ (layers) | 4 |
| $H_{\text{comp}}$ (heads) | 8 |
| $d_{\text{ff}}$ (FFN width) | $4 \times d_{\text{comp}}$ |
| $p$ (dropout) | 0.1 |
| $|\mathcal{Z}|$ (element vocabulary) | 100 |
| Counts $c_t$ (stoichiometry) | Raw (log-count attention bias) |

### A.2. Structural Stream Details

We use the same structural stream across all base architectures unless otherwise noted. Table 8 summarizes shared hyperparameters. We denote the structural stream width $d_{\text{struct}}$, cutoff $r_{\text{cut}}$, radial basis count $K$, mixed channels $K'$, angular cutoff $l_{\text{max}}$, MLP layers $L_{\text{struct}}$, message-passing layers $L_{\text{mp}}$, and cutoff scales $S$.

*Table 8.* Common structural stream hyperparameters used across TriForces models.

| Hyperparameter | Value |
| --- | --- |
| $d_{\text{struct}}$ (width) | 256 |
| $r_{\text{cut}}$ | 6.0 |
| $K$ (radial basis) | 8 |
| $K'$ (mixed channels) | 8 |
| $l_{\max}$ (spherical harmonics) | 4 |
| Radial basis type | Bessel |
| $L_{\text{struct}}$ (MLP layers) | 3 |
| $L_{\text{mp}}$ (message passing) | 2 |
| $S$ (cutoff scales) | 3 (0.5x, 0.75x, 1.0x) |
| Lattice encoding | On (periodic systems) |

**Radial basis and multi-scale envelopes.** We use either a Bessel basis or a Gaussian RBF basis for $\{\phi_k\}_{k=1}^K$, following common atomistic encoders (Schütt et al., 2017; Gasteiger et al., 2020). To improve robustness across local length scales, we employ $S$ smooth cosine cutoffs $\{s_s(r)\}_{s=1}^S$ with different cutoff radii and concatenate the resulting radial features (Gasteiger et al., 2020):

$$\Phi(r) \;=\; \big[s_1(r)\phi_1(r),\ldots,s_1(r)\phi_K(r) \;\|\; \cdots \;\|\; s_S(r)\phi_1(r),\ldots,s_S(r)\phi_K(r)\big]. \tag{14}$$

A learnable linear map then mixes this multi-scale radial representation into $K'$ channels before forming the power spectrum.

**Rotation-invariant power spectrum.** Given coefficients $c_{\alpha lm}(i)$ computed from the mixed radial features, we form invariants $p_{\alpha\alpha'l}(i) = \sum_m c_{\alpha lm}(i)c_{\alpha'lm}(i)$. We retain only the upper triangle $\alpha \leq \alpha'$ to avoid redundancy, and concatenate across $l = 0,\ldots,l_{\max}$. In our implementation we use $l_{\max} = 4$ (i.e., $(l_{\max}+1)^2$ real spherical harmonic components per edge).

**Lattice encoding for periodic systems.** For crystals, we encode the unit cell with a small MLP over nine scale-aware features: normalized lattice vector lengths, normalized inter-vector angles, $\log(\text{volume}/N)$, $\log(\text{density})$, and an orthogonality score. The resulting lattice embedding is broadcast to atoms and concatenated to the invariant local descriptor prior to the MLP projection. We disable this component for non-periodic systems.

**Invariant message passing.** After projecting the invariant descriptor to the stream width, we apply $L_{\text{mp}}$ message passing layers that use only geometric edge features (projected multi-scale radial encodings). Messages are aggregated following Equation (15):

$$\mathbf{h}_j^{(l+1)} = \mathbf{h}_j^{(l)} + \psi^{(l)}\!\left(\left[\mathbf{h}_j^{(l)} \;\Big\|\; \frac{1}{\sum_{i\in\mathcal{N}(j)} s(r_{ij})} \sum_{i\in\mathcal{N}(j)} s(r_{ij})\,\phi^{(l)}\!\Big(\big[\mathbf{h}_i^{(l)} \;\|\; \eta(r_{ij})\big]\Big)\right]\right), \tag{15}$$

where $\eta(r_{ij})$ denotes the learned radial edge embedding obtained by projecting the multi-scale cutoff-weighted radial basis functions introduced in the structural descriptor, $\psi^{(l)}$ is a learned MLP, and $s(r_{ij})$ is the same smooth cutoff envelope used for degree normalization.

### A.3. Stream Fusion Details

We use simple concatenation of stream embeddings at the node level, $\mathbf{h}_i = [\mathbf{h}_i^{\text{comp}} \,\|\, \mathbf{h}_i^{\text{struct}} \,\|\, \mathbf{h}_i^{\text{int}}]$, and analogously for graph-level features. This preserves stream-level interpretability, which is useful for decomposed similarity analysis (Hu et al., 2019; Stärk et al., 2022; Li & Walsh, 2025). More expressive fusion operators (e.g., gated mixing or attention across streams) are possible but not explored here; we leave systematic study of fusion design to future work.

### A.4. Energy and Forces Gradient Coupling

We provide a short derivation clarifying how energy and force supervision interact under conservative training, and state conditions under which adding coordinate-free degrees of freedom like the composition stream reduce this coupling. We focus on the force sensitivity at the first-order, i.e., how much gradient updates that change the energy prediction on a given structure will impact force prediction.

**Notations.** Let $\mathbf{x} = (\mathbf{x}_1,\ldots,\mathbf{x}_N)$, $x_i \in \mathbb{R}^3$ denote atomic positions and let $E_\theta(\mathbf{x},\mathbf{z}) \in \mathbb{R}$ be the predicted total energy,

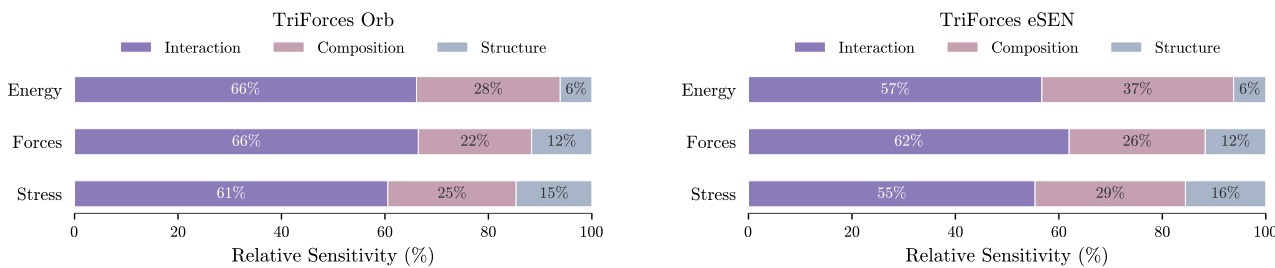

*Figure 6.* Relative sensitivity of predictions to each stream for TriForces Orb-v3 (left) and eSEN (right). Sensitivity is computed as the mean gradient norm of a target scalar (energy sum, force-norm sum, or stress norm) with respect to each stream's node features; higher values indicate stronger dependence. Sensitivity profiles vary by target, indicating complementary roles for composition, structure, and interaction streams.

parametrized by $\theta$, conditioned on species $\mathbf{z}$. Conservative forces are

$$\mathbf{F}_k(\theta) = -\nabla_{\mathbf{x}_k} E_\theta(\mathbf{x}, \mathbf{z}), \qquad k = 1, \ldots, N. \tag{16}$$

Consider the per-structure loss

$$\mathcal{L}(\theta) = (E_\theta - E^*)^2 + \lambda \sum_{k=1}^{N} \|\mathbf{F}_k(\theta) - \mathbf{F}_k^*\|^2. \tag{17}$$

Throughout, we assume that $E_\theta$ is twice continuously differentiable in $(\theta, \mathbf{x})$, which holds with SiLU activation functions, and when the graph neighbours are fixed for a given input, so that mixed partials commute.

**Proposition A.2** (Gradient coupling in conservative training). *Define the energy Jacobian $\boldsymbol{J}_\theta := \frac{\partial E_\theta}{\partial \theta}$ and the mixed Hessians $\boldsymbol{H}_{\theta, \mathbf{x}_k} := \frac{\partial^2 E_\theta}{\partial \theta \, \partial \mathbf{x}_k}$. Then the parameter gradient of (17) is*

$$\nabla_\theta \mathcal{L} = 2(E_\theta - E^*) \boldsymbol{J}_\theta - 2\lambda \sum_{k=1}^{N} (\mathbf{F}_k(\theta) - \mathbf{F}_k^*)^\top \boldsymbol{H}_{\theta, \mathbf{x}_k}^\top. \tag{18}$$

*Proof.* Differentiate the energy term: $\nabla_\theta (E_\theta - E^*)^2 = 2(E_\theta - E^*) \nabla_\theta E_\theta = 2(E_\theta - E^*) \boldsymbol{J}_\theta$. For each force term,

$$\nabla_\theta \|\mathbf{F}_k - \mathbf{F}_k^*\|^2 = 2(\mathbf{F}_k - \mathbf{F}_k^*)^\top \nabla_\theta \mathbf{F}_k. \tag{19}$$

Since $\mathbf{F}_k = -\nabla_{\mathbf{x}_k} E_\theta$,

$$\nabla_\theta \mathbf{F}_k = -\nabla_\theta \nabla_{\mathbf{x}_k} E_\theta = -\nabla_{\mathbf{x}_k} \nabla_\theta E_\theta = -\left( \frac{\partial^2 E_\theta}{\partial \theta \, \partial \mathbf{x}_k} \right)^\top = -\boldsymbol{H}_{\theta, \mathbf{x}_k}^\top, \tag{20}$$

where we used commutativity of mixed partials. Substituting into (17) yields (18). $\square$

**Force-preserving directions.** For a small parameter update $\Delta\theta$, a Taylor expansion gives

$$\mathbf{F}_k(\theta + \Delta\theta) = \mathbf{F}_k(\theta) + \nabla_\theta \mathbf{F}_k(\theta) \Delta\theta + o(\|\Delta\theta\|), \tag{21}$$

hence

$$\delta\mathbf{F}_k := \mathbf{F}_k(\theta + \Delta\theta) - \mathbf{F}_k(\theta) = \nabla_\theta \mathbf{F}_k(\theta) \Delta\theta + o(\|\Delta\theta\|). \tag{22}$$

Using $\nabla_\theta \mathbf{F}_k(\theta) = -\boldsymbol{H}_{\theta, \mathbf{x}_k}^\top$ (from Proposition A.2),

$$\delta\mathbf{F}_k = -\boldsymbol{H}_{\theta, \mathbf{x}_k}^\top \Delta\theta + o(\|\Delta\theta\|). \tag{23}$$

Based on this, let the *force-preserving subspace* (at first-order) be

$$\mathcal{S}_\theta := \bigcap_{k=1}^{N} \mathrm{Null}\left( \boldsymbol{H}_{\theta, \mathbf{x}_k}^\top \right), \tag{24}$$

so that $\Delta\theta \in \mathcal{S}_\theta$ implies $\delta\mathbf{F}_k = o(\|\Delta\theta\|)$ for all $k$.

**Energy descent with force preservation.** Let $\mathbf{g}_E := \nabla_\theta E_\theta(\mathbf{x}, \mathbf{z}) = \boldsymbol{J}_\theta$. Any update direction $\Delta\theta$ that preserves forces to first order must satisfy $\Delta\theta \in \mathcal{S}_\theta$. Consequently, among force-preserving directions, the energy decrease is governed by the projection of $\mathbf{g}_E$ onto $\mathcal{S}_\theta$:

$$E_\theta(\theta + \Delta\theta) - E_\theta(\theta) = \mathbf{g}_E^\top \Delta\theta + o(\|\Delta\theta\|), \qquad \Delta\theta \in \mathcal{S}_\theta. \tag{25}$$

This shows that energy and force trade-offs can arise when $\mathbf{g}_E$ has limited component in $\mathcal{S}_\theta$.

**Lemma A.3** (Force invariance under additive energy terms). *Suppose the energy decomposes as*

$$E_\theta(\mathbf{x}, \mathbf{z}) = E_{geom}(\mathbf{x}, \mathbf{z}; \theta_{geom}) + E_{cf}(\mathbf{z}; \theta_{cf}), \tag{26}$$

*where $E_{cf}$ does not depend on $\mathbf{x}$. Then for all $k$, $\nabla_{\mathbf{x}_k} E_{cf}(\mathbf{z}; \theta_{cf}) = \mathbf{0}$ and hence*

$$\mathbf{F}_k(\theta) = -\nabla_{\mathbf{x}_k} E_{geom}(\mathbf{x}, \mathbf{z}; \theta_{geom}), \tag{27}$$

*so any update to $\theta_{cf}$ can change energy while leaving forces exactly unchanged. Equivalently, $\boldsymbol{H}_{\theta_{cf}, \mathbf{x}_k} = \mathbf{0}$ for all $k$.*

*Proof.* Because $E_{\mathrm{cf}}$ is independent of $\mathbf{x}$, its partial derivative with respect to any $\mathbf{x}_k$ is zero. Therefore the forces depend only on $E_{\mathrm{geom}}$, and mixed derivatives $\frac{\partial^2 E_{\mathrm{cf}}}{(\partial\theta_{\mathrm{cf}}\partial\mathbf{x}_k)}$ vanish identically. $\qquad\square$

**Proposition A.4** (Rank bound for a one-hidden-layer MLP head). *Let the energy head take concatenated inputs $\mathbf{h} = [\mathbf{h}^{comp}\|\mathbf{h}^{geom}]$ with $\mathbf{h}^{comp} \in \mathbb{R}^{d_c}$ and $\mathbf{h}^{geom} \in \mathbb{R}^{d_g}$, and define*

$$E(\mathbf{h}^{comp}, \mathbf{h}^{geom}) = \mathbf{v}^\top \sigma(\mathbf{a}), \qquad \mathbf{a} = \mathbf{W}_c \mathbf{h}^{comp} + \mathbf{W}_g \mathbf{h}^{geom} + \mathbf{b}, \tag{28}$$

*where $\sigma$ is applied elementwise, $\mathbf{W}_c \in \mathbb{R}^{m\times d_c}$, $\mathbf{W}_g \in \mathbb{R}^{m\times d_g}$, $\mathbf{v}, \mathbf{b} \in \mathbb{R}^m$, and $m$ is the hidden width. Assume $\sigma$ is twice differentiable. Then the cross second derivative gives*

$$\frac{\partial^2 E}{\partial\mathbf{h}^{comp}\,\partial\mathbf{h}^{geom}} = \mathbf{W}_c^\top \mathbf{D}(\mathbf{a})\mathbf{W}_g, \qquad \mathbf{D}(\mathbf{a}) = \mathrm{Diag}(\mathbf{v} \odot \sigma''(\mathbf{a})), \tag{29}$$

*and satisfies the rank bound*

$$\mathrm{rank}\left(\frac{\partial^2 E}{\partial\mathbf{h}^{comp}\,\partial\mathbf{h}^{geom}}\right) \leq \min\{\mathrm{rank}(\mathbf{W}_c), \mathrm{rank}(\mathbf{W}_g), m\}. \tag{30}$$

*Proof.* Applying the chain rule,

$$\frac{\partial E}{\partial\mathbf{h}^{comp}} = \mathbf{W}_c^\top(\mathbf{v} \odot \sigma'(\mathbf{a})).$$

Differentiating with respect to $\mathbf{h}^{geom}$ yields

$$\frac{\partial^2 E}{\partial\mathbf{h}^{comp}\,\partial\mathbf{h}^{geom}} = \mathbf{W}_c^\top \mathrm{Diag}(\mathbf{v} \odot \sigma''(\mathbf{a}))\mathbf{W}_g.$$

The rank bound follows from $\mathrm{rank}(\mathbf{ABC}) \leq \min\{\mathrm{rank}(\mathbf{A}), \mathrm{rank}(\mathbf{B}), \mathrm{rank}(\mathbf{C})\}$. $\qquad\square$

**Proposition A.5** (Concatenated streams on force-preserving subspace). *Let $\theta_{\mathrm{comp}}$ denote parameters that influence the energy only through $\mathbf{h}^{comp}$, and define the stacked mixed Hessian*

$$\widetilde{\boldsymbol{H}}_{\mathrm{comp}} = \begin{bmatrix} \frac{\partial^2 E}{\partial\theta_{\mathrm{comp}}\,\partial\mathbf{x}_1}^\top \\ \vdots \\ \frac{\partial^2 E}{\partial\theta_{\mathrm{comp}}\,\partial\mathbf{x}_N}^\top \end{bmatrix}.$$

*If the feature-level Hessian $\frac{\partial^2 E}{\partial\mathbf{h}^{comp}\,\partial\mathbf{h}^{geom}}$ has rank at most $r$, then*

$$\mathrm{rank}(\widetilde{\boldsymbol{H}}_{\mathrm{comp}}) \leq r, \qquad \dim(\mathrm{Null}(\widetilde{\boldsymbol{H}}_{\mathrm{comp}})) \geq \dim(\theta_{\mathrm{comp}}) - r. \tag{31}$$

*Proof.* By the chain rule,

$$\forall\, k \in [1, N], \quad \frac{\partial^2 E}{\partial \theta_{\text{comp}}\, \partial \mathbf{x}_k} = \left( \frac{\partial \mathbf{h}^{\text{comp}}}{\partial \theta_{\text{comp}}} \right)^{\top} \frac{\partial^2 E}{\partial \mathbf{h}^{\text{comp}}\, \partial \mathbf{h}^{\text{geom}}} \frac{\partial \mathbf{h}^{\text{geom}}}{\partial \mathbf{x}_k}.$$

Each block is therefore a product of matrices whose middle factor has rank at most $r$. This is true for all $k$, hence $\text{rank}(\widetilde{\boldsymbol{H}}_{\text{comp}}) \leq r$. The kernel bound follows from rank–nullity. $\qquad\square$

**Application to TriForces.** TriForces introduces a composition stream $\mathbf{h}^{\text{comp}}$ that is coordinate-free by construction. Lemma A.3 identifies a sufficient mechanism where such a stream can reduce energy and force coupling: if the downstream energy head learns an approximately additive coordinate-free component of the form (26), then the associated parameters admit force-preserving update directions.

While this additive structure is not guaranteed, the TriForces architecture biases optimization toward such solutions by (i) explicitly exposing coordinate-free features through $\mathbf{h}^{\text{comp}}$, and (ii) using a shallow fusion head, which empirically exhibits low cross-sensitivity between compositional and geometric features. More generally, for an arbitrary fusion head, the influence of composition parameters on forces is governed by the stacked mixed Hessian

$$\widetilde{\boldsymbol{H}}_{\text{comp}} = \begin{bmatrix} \boldsymbol{H}_{\theta_{\text{comp}}, \mathbf{x}_1}^{\top} \\ \vdots \\ \boldsymbol{H}_{\theta_{\text{comp}}, \mathbf{x}_N}^{\top} \end{bmatrix}.$$

Smaller operator norm or lower effective rank of $\widetilde{\boldsymbol{H}}_{\text{comp}}$ implies a larger force-preserving subspace via (23). This rank is bounded by the MLP hidden width and depends on both the learned decoder weights and the input.

## B. Additional Experiments

### B.1. MLIPs

Figure 7 shows OMat24 validation curves for energy, forces, and stress. TriForces-Streams reduces energy error early without impacting forces learning, while full TriForces improves all three metrics throughout training and transfers these gains to a validation split curated from MPtrj.

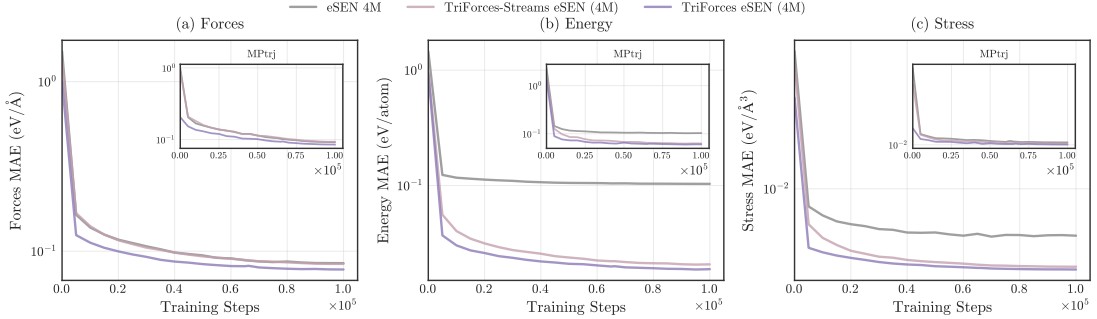

*Figure 7.* Validation curves during eSEN fine-tuning on 4M OMat24 samples, evaluated on OMat24 val and a 10K MPtrj split (inset). Curves show energy, force, and stress MAE over training. TriForces-Streams lowers energy early and full TriForces improves all three metrics, with gains transferring to MPtrj.

### B.2. Data Efficiency

Table 9 reports fine-tuning MAE of direct eSEN variants across dataset sizes. TriForces yields the largest relative gains in the low-data regime (20K to 100K samples) and maintains improvements as data increases, indicating better sample efficiency rather than only higher capacity. All the experiments were run following the hyperparameters in App. 20, but for 50K iteration steps.

*Table 9.* Data efficiency on OMat24 for direct eSEN variants across dataset sizes (20K–2M samples). Energy and force MAE (↓) are reported; **Bold** = best and underline = second best. TriForces achieves the lowest errors at every size, with the largest relative gains in the 20K–100K sample regime.

| Samples | Energy MAE (meV/atom) | | | Forces MAE (meV/Å) | | |
|---|---|---|---|---|---|---|
| | eSEN Base | + TriForces-Streams | TriForces | eSEN Base | + TriForces-Streams | TriForces |
| 20K | 81.3 | 42.8 | **34.6** | 151.4 | 145.8 | **121.8** |
| 100K | 57.5 | 25.7 | **23.0** | 107.4 | 107.0 | **101.3** |
| 500K | 54.0 | 22.0 | **21.2** | 100.8 | 99.7 | **97.7** |
| 1M | 52.7 | 21.0 | **20.9** | 99.2 | 98.3 | **97.4** |
| 2M | 53.0 | 20.8 | **20.8** | 98.8 | 97.9 | **97.2** |

## B.3. SSL vs. Supervised Pretraining

Table 10 compares pretraining objectives and pretraining-data scale before downstream OMat24 fine-tuning. In this controlled setting, SSL pretraining gives a better downstream initialization than supervised pretraining on LeMat-Bulk, especially for forces. Increasing unlabeled OMat pretraining data also improves downstream performance, indicating that representation quality continues to benefit from scale.

*Table 10.* Effect of pretraining objective and scale on downstream OMat24 fine-tuning. Energy is reported in meV/atom and forces in meV/Å.

| Pretraining | Orb $E \downarrow$ | Orb $F \downarrow$ | eSEN $E \downarrow$ | eSEN $F \downarrow$ |
|---|---|---|---|---|
| Supervised LeMat-Bulk | – | – | 22.5 | 106.0 |
| SSL LeMat-Bulk | – | – | 18.8 | 86.8 |
| SSL OMat-1M | 21.9 | 112.5 | 21.1 | 94.2 |
| SSL OMat-10M | 19.8 | 103.6 | 20.4 | 93.8 |
| SSL OMat-100M | 18.9 | 99.5 | 18.7 | 87.1 |

## B.4. MatBench Discovery

Table 11 reports metrics for both the unique-prototype and full test sets. TriForces Orb-v3 improves MAE and RMSD relative to Orb-v3 across splits, while other metrics remain comparable. TriForces eSEN-sm remains close to the larger eSEN-30M-OAM on most metrics; MAE and RMSD differ by about 0.001 (eV/atom and Å, respectively) and $\kappa_{SRME}$ is slightly higher (0.206 vs 0.170). Our TriForces Orb-v3 used a maximum neighbour count of 40 versus 120 for Orb-v3

## B.5. QM9

Table 12 shows the complete QM9 results. Pretraining on OMol yields the lowest MAE across all reported targets, while bulk-only pretraining provides smaller gains, indicating that molecular pretraining data is important for QM9 transfer.

## B.6. Full Ablation on Orb-v3

This appendix provides complete ablation results summarized in the main text. Table 13 shows all pretraining configurations tested on Orb-v3, pretrained for 100K steps on bulks and fine-tuned on OMat24. The full table confirms the trends in the main ablation: removing compositional or structural streams degrades the corresponding probes and downstream energy/force metrics, and LeJEPA-only training performs poorly on fine-tuning. Denoising is most impactful for non-equivariant Orb-v3, while masking provides smaller but consistent gains. The OMat24 and MPtrj columns track each other closely, indicating that the ablation effects are consistent across transfer targets.

## B.7. Parameter-matched Baselines

To rule out that TriForces gains are simply from additional parameters, we compare against *plain single-stream* models with matched capacity by widening the interaction stream to match the TriForces parameter count (within a small tolerance), while keeping data, objectives, and training schedules fixed. For eSEN, we double edge channels and spherical channels; for Orb, we set the latent dimension to 512 and increase message passing steps to 6 (from 5). These parameter-matched

*Table 11.* MatBench Discovery full metrics for unique-prototype and full test sets. Prec: precision; F1: F1 score; DAF: discovery acceleration factor; Acc: accuracy; MAE: mean absolute error (eV/atom); $R^2$: coefficient of determination; $\kappa_{\text{SRME}}$: thermal conductivity scaled relative mean error; RMSD: root mean square displacement (Å). TriForces Orb-v3 improves MAE/RMSD vs Orb-v3 on both splits, and TriForces eSEN-sm remains close to eSEN-30M-OAM despite being much smaller. Note: TriForces Orb-v3 uses max neighbors=40 vs 120 for Orb-v3, so $\kappa_{\text{SRME}}$ is not directly comparable.

| Model | Test Set | Prec | F1 | DAF | Acc | MAE | $R^2$ | $\kappa$SRME | RMSD |
|---|---|---|---|---|---|---|---|---|---|
| eSEN-30M-OAM | Unique | 0.977 | 0.925 | 6.07 | 0.977 | 0.018 | 0.866 | 0.170 | 0.061 |
| | Full | 0.967 | 0.902 | 5.28 | 0.967 | 0.018 | 0.860 | 0.170 | 0.061 |
| TriForces eSEN-sm | Unique | 0.909 | 0.915 | 5.94 | 0.974 | 0.019 | 0.874 | 0.206 | 0.062 |
| | Full | 0.882 | 0.894 | 5.29 | 0.964 | 0.019 | 0.861 | 0.206 | 0.062 |
| Orb-v3 | Unique | 0.971 | 0.905 | 5.91 | 0.971 | 0.024 | 0.821 | 0.210 | 0.075 |
| | Full | 0.961 | 0.887 | 5.16 | 0.961 | 0.023 | 0.820 | 0.210 | 0.075 |
| TriForces Orb-v3 | Unique | 0.903 | 0.910 | 5.91 | 0.972 | 0.020 | 0.874 | 0.460 | 0.065 |
| | Full | 0.877 | 0.889 | 5.26 | 0.962 | 0.020 | 0.861 | 0.460 | 0.064 |

*Table 12.* Full QM9 results on all 12 targets. eSEN: no SSL pretraining. TriForces eSEN (Bulk): TriForces pretrained on crystal bulks. TriForces eSEN (OMol): TriForces pretrained on mix of bulks, slabs, and molecules. MAE ($\downarrow$). **Bold**: best; underline: second best.

| **Model** | $\mu$ (D) | $\alpha$ ($a_0^3$) | $\varepsilon_{\text{HOMO}}$ (meV) | $\varepsilon_{\text{LUMO}}$ (meV) | $\Delta\varepsilon$ (meV) | $\langle R^2 \rangle$ ($a_0^2$) | ZPVE (meV) | $U_0$ (meV) | $U$ (meV) | $H$ (meV) | $G$ (meV) | $C_v$ ($\frac{\text{cal}}{\text{mol K}}$) |
|---|---|---|---|---|---|---|---|---|---|---|---|---|
| eSEN | 0.023 | 0.098 | 23.6 | 25.1 | 45.1 | 0.750 | 3.49 | 9.0 | **9.0** | 9.0 | 9.5 | 0.037 |
| TriForces eSEN (Bulk) | 0.021 | 0.093 | 23.7 | 25.2 | 47.3 | 0.754 | 3.61 | 9.3 | 9.3 | 9.4 | 9.7 | 0.041 |
| TriForces eSEN (All) | 0.018 | 0.079 | 21.0 | 21.0 | 40.0 | **0.540** | **2.72** | **7.6** | 9.1 | **7.8** | 8.3 | 0.032 |
| TriForces eSEN (OMol) | **0.018** | **0.075** | **20.2** | **20.4** | **38.8** | 0.545 | 2.73 | 8.9 | **9.0** | 9.0 | **8.0** | **0.031** |

baselines are more expensive per step because added capacity is placed in the interaction stream. We report MatBench and QM9 results for eSEN and Orb in the tables below; TriForces remains noticeably better, indicating improvements are not solely due to parameter count.

# C. Analysis of Learned Representations

## C.1. Probing

To assess whether pretrained representations retain fundamental structural and compositional information, we evaluate linear probing on frozen embeddings for four diagnostic tasks that capture basic geometric and chemical properties:

- **Crystal system** (7-class): The symmetry category of the crystal lattice (e.g., cubic, hexagonal, tetragonal), determined purely by geometry.
- **Majority element** (118-class): The most frequently occurring element in the structure's composition.
- **Coordination number**: The average number of nearest neighbors per atom, calculated using the effective coordination number (ECoN) (Hoppe, 1979).
- **Mean nearest-neighbor distance**: The average distance to each atom's closest neighbor in Ångstroms, capturing local atomic spacing.

These tasks require no DFT labels and test whether basic geometric and chemical information remains accessible in the learned representations.

Figure 8 reveals that baseline MLIPs struggle on these seemingly simple tasks despite being pretrained on over 100M structures with only at most 62% accuracy in retrieving a seemingly simple property as the maximal chemical element. This indicates that supervised training on energies and forces discards or hides information not directly needed for those targets, but needed for describing a chemical system.

TriForces variants achieve near-perfect classification (96–100%) and substantially lower regression error. The mean NN distance MAE improves from 0.75–1.10 Å to 0.05–0.08 Å. These results confirm that the three-stream architecture preserves compositional and structural factors, making them directly accessible.

*Table 13.* Full ablation study on Orb-v3 with all metrics. See Sec. 4.6 for discussion.

| Model | Streams | | SSL | | | Probing | | | | Fine-tuning | | | | |
|---|---|---|---|---|---|---|---|---|---|---|---|---|---|---|
| | C | S | D | M | J | $d_{\text{NN}}$ | $E_f$ | Cry | Uni | $E_O$ | $F_O$ | $\sigma_O$ | $E_M$ | $F_M$ |
| TriForces | ✓ | ✓ | ✓ | ✓ | ✓ | 0.026 | 0.121 | 61.7 | -2.6 | 0.023 | 0.092 | 0.0047 | 0.062 | 0.084 |
| w/o Comp | ✗ | ✓ | ✓ | ✓ | ✓ | 0.030 | 0.152 | 68.5 | -2.3 | 0.026 | 0.112 | 0.0051 | 0.091 | 0.122 |
| w/o Struct | ✓ | ✗ | ✓ | ✓ | ✓ | 0.034 | 0.116 | 59.0 | -0.9 | 0.022 | 0.092 | 0.0047 | 0.062 | 0.083 |
| w/o Streams | ✗ | ✗ | ✓ | ✓ | ✓ | 0.035 | 0.162 | 57.2 | -1.2 | 0.095 | 0.098 | 0.0063 | 0.101 | 0.089 |
| w/o LeJEPA | ✓ | ✓ | ✓ | ✓ | ✗ | 0.029 | 0.108 | 59.8 | -0.6 | 0.022 | 0.094 | 0.0048 | 0.063 | 0.087 |
| w/o Mask | ✓ | ✓ | ✓ | ✗ | ✓ | 0.030 | 0.129 | 71.7 | -1.9 | 0.020 | 0.101 | 0.0049 | 0.080 | 0.108 |
| LeJEPA only | ✓ | ✓ | ✗ | ✗ | ✓ | 0.038 | 0.157 | 68.4 | -2.0 | 0.033 | 0.156 | 0.0061 | 0.118 | 0.162 |
| Barlow | ✓ | ✓ | ✓ | ✓ | BT | 0.027 | 0.115 | 75.6 | -2.3 | 0.022 | 0.098 | 0.0048 | 0.073 | 0.102 |
| Crystal Twins | ✓ | ✓ | ✓ | ✓ | CT | 0.055 | 0.304 | 57.7 | -0.1 | 0.043 | 0.181 | 0.0069 | 0.117 | 0.208 |
| De-Noise | ✗ | ✗ | Z | ✗ | ✗ | 0.038 | 0.144 | 55.6 | -0.9 | 0.095 | 0.095 | 0.0065 | 0.102 | 0.094 |
| LeMat-Traj | ✓ | ✓ | ✓ | ✓ | ✓ | 0.031 | 0.123 | 68.2 | -2.0 | 0.020 | 0.102 | 0.0049 | 0.079 | 0.111 |

C/S/D/M/J = Comp/Struct/Denoise/Mask/JEPA. $d_{\text{NN}}$: nearest neighbor distance (Å). $E_f$: formation energy (eV/at). Cry: crystal system (%). Uni: uniformity (Wang & Isola, 2020). $E/F/\sigma$: energy/force/stress MAE. Subscripts O/M = OMAT/MPTRJ. BT = Barlow Twins, Z = Zaidi.

*Table 14.* Parameter-matched comparison: TriForces models vs. base models with matched parameter count. MAE (↓) / F1 (↑), fold 0 only. **Bold**: best within backbone.

| Task (Units) | eSEN (Matched) | TriForces eSEN | Orb (Matched) | TriForces Orb |
|---|---|---|---|---|
| Phonons ($cm^{-1}$) | 33.3 | **22.4** | 34.6 | **30.7** |
| Dielectric | 0.157 | **0.122** | 0.174 | **0.126** |
| Log GVRH ($\log_{10}$(GPa)) | 0.074 | **0.058** | 0.078 | **0.057** |
| Log KVRH ($\log_{10}$(GPa)) | 0.048 | **0.042** | 0.057 | **0.043** |
| Perovskites (meV) | 33.2 | **25.8** | 32.0 | **26.0** |
| MP Gap (eV) | 0.309 | **0.159** | 0.352 | **0.131** |
| MP E Form (meV/atom) | 35.8 | **20.3** | 27.2 | **17.2** |
| MP Is Metal (F1) | 0.831 | **0.856** | 0.822 | **0.903** |

## C.2. Latent Space Visualization

Figure 9 illustrates the resulting separation of chemistry and structure in the embedding space.

## C.3. Qualitative Retrieval Comparison

We visualize nearest neighbors for query structures using TriForces and baseline embeddings. TriForces yields interpretable neighborhoods where composition- and structure-based similarity are separated, while baseline embeddings trained only on energies and forces tend to conflate the two factors. Figure 10 shows a representative example.

Table 17 provides representative structure pairs that illustrate the decomposition. Same-skeleton pairs share geometry but differ in composition, polymorph pairs separate structural from compositional similarity, and StructureMatcher confirms whether geometries are equivalent despite different reported space groups.

## D. Training Details

### D.1. Pipeline Summary and Pseudocode

Table 18 summarizes the training pipelines used for the main model families and clarifies whether improvements come from random-initialized stream factorization, SSL initialization, or downstream fine-tuning. Algorithm 1 summarizes the TriForces forward pass for two augmented views and the SSL objectives applied to the resulting stream embeddings.

*Table 15.* Parameter-matched comparison on QM9: TriForces eSEN vs. eSEN with matched parameter count. MAE (↓). **Bold**: best.

| Model | $\mu$ (D) | $\alpha$ ($a_0^3$) | $\varepsilon_{\text{HOMO}}$ (meV) | $\varepsilon_{\text{LUMO}}$ (meV) | $\Delta\varepsilon$ (meV) | $C_v$ ($\frac{\text{cal}}{\text{mol K}}$) | $U_0$ (meV) |
|---|---|---|---|---|---|---|---|
| eSEN (Matched) | 0.020 | 0.091 | 22.4 | 23.1 | 41.9 | 0.032 | **8.3** |
| TriForces eSEN | **0.018** | **0.075** | **20.2** | **20.4** | **38.8** | **0.031** | 8.9 |

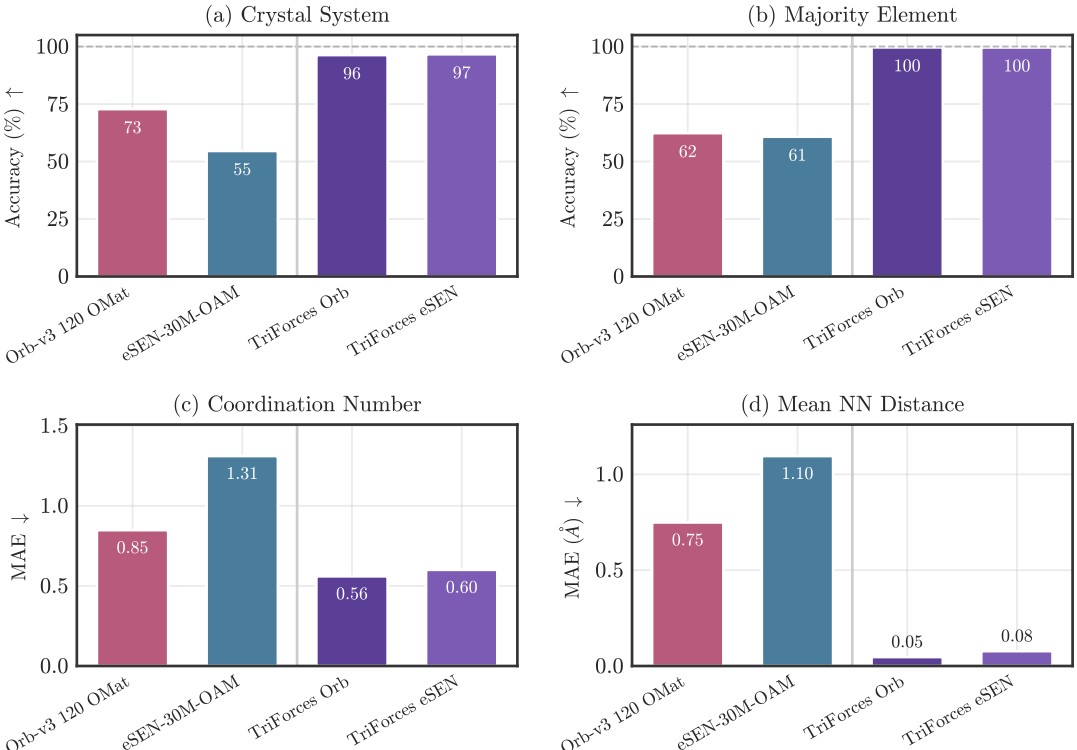

*Figure 8.* Probing on frozen embeddings with MLP heads for (a) crystal system, (b) majority element, (c) coordination number, and (d) mean nearest-neighbor distance. Accuracies are shown for (a,b) and MAE for (c,d). Fully trained MLIP variants reach only 55–73% and 61–62% accuracy, while TriForces reaches 96–100% and reduces coordination/NN-distance MAE by 34%/94%.

## D.2. Computational Cost

Figures 11 and 12 summarize the computational overhead of adding TriForces streams to different base architectures. We report inference (forward) time, training (forward+backward) time, and peak GPU memory as a function of the number of atoms. TriForces introduces a moderate constant-factor overhead at small system sizes, which decreases with system size as compute becomes dominated by the interaction stream's message passing. This also clarifies why uniformly scaling the backbone to match TriForces' total parameter count can be more expensive: additional parameters in the interaction stream directly increase the dominant compute path, whereas TriForces allocates a non-trivial fraction of parameters to auxiliary streams that do not scale as steeply with system size. All timings are measured in **FP32** on a single NVIDIA A100-80GB GPU; missing points correspond to out-of-memory (OOM) measurements.

## D.3. Pretraining Hyperparameters

**Training time.** Pretraining TriForces-eSEN for 300k steps (∼5 epochs) on 1 H100 GPU with batch size 512 in bfloat16 mixed precision takes approximately 40 hours. TriForces-Orb takes ∼20 hours on 1 H100 GPU (∼20 GPU-hours). Ablation pretraining runs used 100k steps (∼5 epochs) with the same batch size and precision. We generate two augmented views per structure for SSL objectives, and pretraining uses no DFT labels. All pretraining variants use identical hyperparameters;

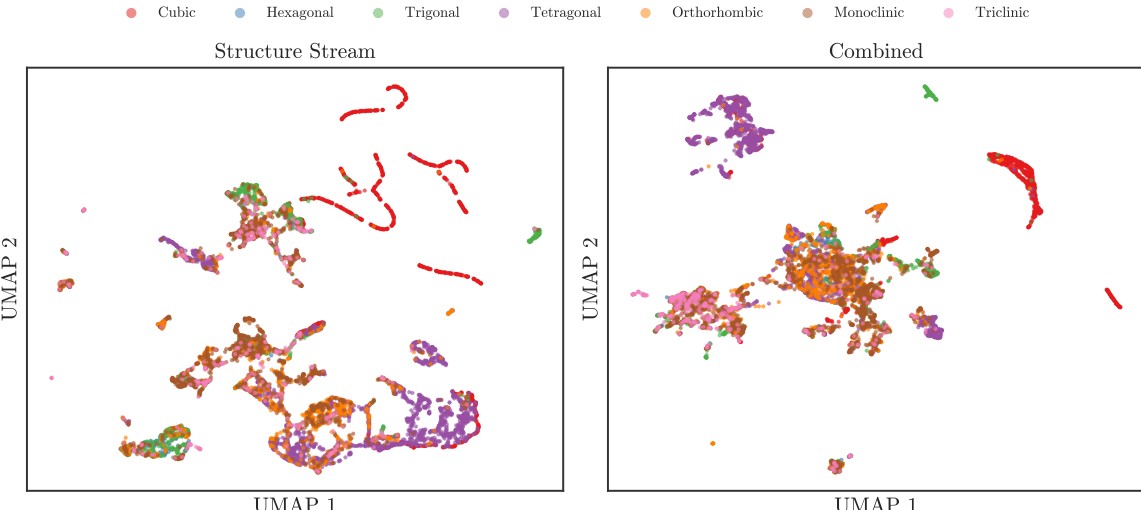

*Figure 9.* UMAP visualization of embeddings. Single-stream models entangle composition and structure, while TriForces separates them: $\mathbf{h}^{comp}$ clusters by chemistry and $\mathbf{h}^{struct}$ clusters by crystal system. Stream decomposition yields a more interpretable latent structure.

*Table 16.* Retrieval baselines at R@10. TriForces uses the composition stream for element-set retrieval and the structure stream for space-group retrieval.

| Method | Element Set R@10 | Space Group R@10 |
|---|---|---|
| SOAP | 0.814 | 0.103 |
| SineMatrix | 0.041 | 0.221 |
| eSEN 30M OAM | 0.184 | 0.207 |
| Orb-v3 OMat | 0.179 | 0.413 |
| TriForces eSEN | 0.770 | 0.648 |
| TriForces Orb-v3 | 0.774 | 0.642 |

only the dataset changes (LeMat-Bulk, OMol25, or their union). Final models use 300k steps (∼5 epochs); ablations use 100k steps (∼5 epochs). Table 19 lists the shared pretraining hyperparameters.

### D.4. Fine-tuning Hyperparameters

**OMat24 fine-tuning compute.** Fine-tuning for Table 1 uses ∼24 GPU-hours on a single A100 GPU in float32.

**Task heads.** For all targets, we use a direct prediction head with 2 hidden layers of width 128, SiLU activations, and no normalization. For energy-conserving models, we use the same head to predict energy, with forces obtained as its positional gradients.

**MatBench and QM9.** For MatBench we fine-tune for 50,000 steps, and for QM9 we fine-tune for 50 epochs.

Models for the Matbench Discovery were trained following in parts the pipeline established in Fu et al. (2025) without using the direct training warm-up. Table 21 shows the different hyperparameters for these models. We used 4 A100 GPUs; total fine-tuning cost was 1500 GPU-hours for TriForces-eSEN and 1200 GPU-hours for TriForces-Orb. SSL pre-training consumed 35 H100 GPU-hours for eSEN and 30 H100 GPU-hours for Orb (bfloat16 mixed precision).

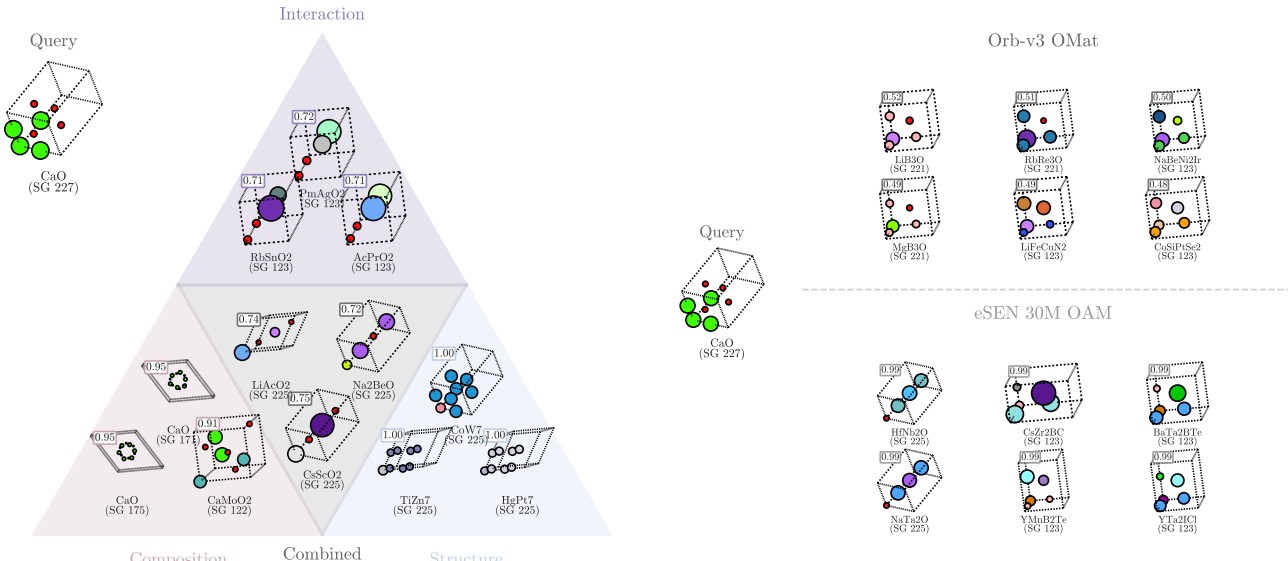

*Figure 10.* Qualitative nearest-neighbor retrieval for a query: TriForces streams (left) vs. single-stream MLIPs trained on energies and forces (right). For each stream, the cosine similarity between the two embeddings is reported. TriForces yields neighborhoods that separate composition and structure-based similarity, whereas candidates and similarity scores from baselines are not interpretable and qualitatively different from the input.

*Table 17.* Structure-pair examples demonstrating disentangled embeddings. Columns report cosine similarity from the composition stream (Comp.), structural stream (Struct.), combined embedding, and baseline embedding; SM reports StructureMatcher (Ong et al., 2013) equivalence. TriForces assigns high Struct. to same-skeleton pairs and high Comp. to polymorphs, separating geometry from chemistry.

| Group | Structure 1 | Structure 2 | Comp. | Struct. | Combined | Baseline | SM |
|---|---|---|---|---|---|---|---|
| Same Skeleton | | | 0.31 | **1.00** | 0.74 | 0.48 | No |
| Polymorphs (SM=No) | | | **1.00** | -0.04 | 0.76 | 0.09 | No |
| Polymorphs (SM=Yes) | | | **1.00** | **0.31** | 0.96 | 0.83 | Yes (0.00Å) |

*Table 18.* Summary of training pipelines for the main comparisons.

| Model/setting | Initialization | Pretraining data | Pretraining objective | Fine-tuning data | Fine-tuning budget |
|---|---|---|---|---|---|
| Baseline MLIP | Random init. | – | – | OMat24 4M | 100k steps / 2 epochs |
| TriForces-Streams | Random init. | – | – | OMat24 4M | 100k steps / 2 epochs |
| TriForces | SSL init. | LeMat-Bulk / OMol25 / OMat variants | Denoise + mask + LeJEPA | OMat24 4M | 100k steps / 2 epochs |
| MatBench transfer | SSL or DFT init. | LeMat-Bulk or DFT-pretrained checkpoints | SSL or supervised labels | MatBench folds | 50k steps |
| QM9 transfer | SSL init. | Bulk, OMol25, or both | Denoise + mask + LeJEPA | QM9 split | 50 epochs |
| MatBench Discovery | SSL init. | LeMat-Bulk | Denoise + mask + LeJEPA | OMat24, then MPtrj+sAlex | 400k + 400k steps |

---

**Algorithm 1** TriForces forward pass and self-supervised pretraining step

---

1: Initialize the interaction GNN $f_{\text{int}}$, composition encoder $f_{\text{comp}}$, and structure encoder $f_{\text{struct}}$.
2: **for** each input structure $\mathcal{G} = (Z, \mathbf{x}, \mathcal{E})$ **do**
3:     Sample two stochastic views $\tilde{\mathcal{G}}_1, \tilde{\mathcal{G}}_2$ using atom masking, position noise, graph perturbations, and rotations when applicable.
4:     **for** $v \in \{1, 2\}$ **do**
5:         Compute interaction features $\mathbf{h}_v^{\text{int}} = f_{\text{int}}(\tilde{Z}_v, \tilde{\mathbf{x}}_v, \tilde{\mathcal{E}}_v)$.
6:         Compute composition features $\mathbf{h}_v^{\text{comp}} = f_{\text{comp}}(\tilde{Z}_v)$ from element identities and counts.
7:         Compute structure features $\mathbf{h}_v^{\text{struct}} = f_{\text{struct}}(\tilde{\mathbf{x}}_v, \tilde{\mathcal{E}}_v)$ from invariant geometric features.
8:         Concatenate the streams, $\mathbf{h}_v = [\mathbf{h}_v^{\text{int}}; \mathbf{h}_v^{\text{comp}}; \mathbf{h}_v^{\text{struct}}]$.
9:     **end for**
10:     Apply denoising (Eq. 11) and masking (Eq. 12) losses to recover position perturbations and masked atom types.
11:     Apply node-level and graph-level LeJEPA losses between $\mathbf{h}_1$ and $\mathbf{h}_2$ (Eq. 9).
12:     Update all streams with the combined SSL objective in Eq. 13.
13: **end for**

---

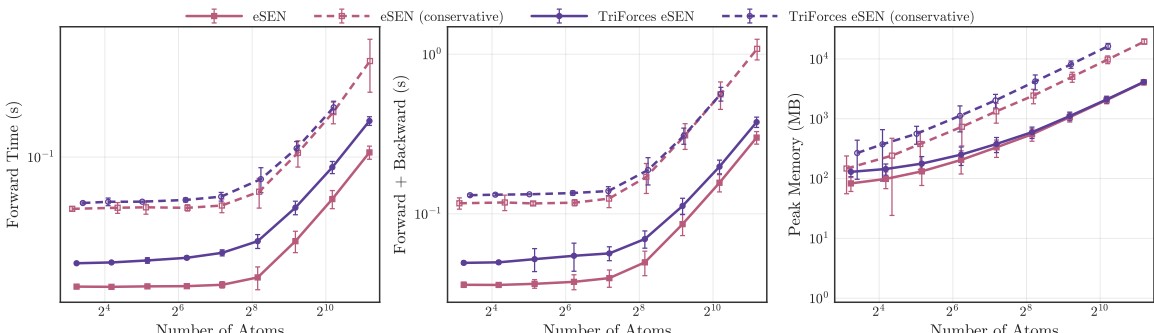

*Figure 11.* Computational scaling comparison between TriForces eSEN (12M parameters) and the interaction eSEN stream (6M parameters). Solid lines show the direct model, dashed lines show the conservative model. TriForces adds  25% training overhead sizes with comparable scaling.

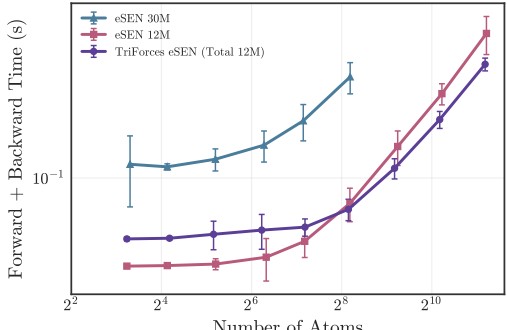

*Figure 12.* Forward + backward time of TriForces eSEN against larger base eSEN models. On larger systems, TriForces eSEN 12M ($l_{\max} = 2$) trains 1.35x faster than the parameter-matched eSEN 12M ($l_{\max} = 2$) baseline and 4.7x faster than eSEN 30M ($l_{\max} = 3$).

*Table 19.* Pretraining hyperparameters (shared across dataset variants).

| Hyperparameter | Value |
|---|---|
| *Training* | |
| Steps | 300,000 |
| Epochs | 5 |
| Effective batch size | 512 |
| Optimizer | AdamW |
| Learning rate | $3 \times 10^{-4}$ |
| LR schedule | Linear warmup + cosine decay |
| Warmup steps | 500 |
| Weight decay | 0.01 |
| Gradient clipping | 10.0 |
| Precision | bfloat16 (mixed) |
| *Loss weights* | |
| $\lambda_{\text{denoise}}$ | 10 |
| $\lambda_{\text{atom}}$ (masking) | 0.1 |
| $\lambda_{\text{LeJEPA}}^{\text{node}}$ | 0.1 |
| $\lambda_{\text{LeJEPA}}^{\text{graph}}$ | 0.1 |
| $\lambda_{\text{SIGReg}}$ | 0.1 |
| *LeJEPA* | |
| Number of augmentations | 2 |
| SIGReg slices | 1024 |
| SIGReg $t_{\max}$ | 3.0 |
| Quadrature points | 17 |
| *Augmentations* | |
| Max augmentations per sample | 3 |
| Noise | Always applied |
| Noise $\sigma$ range | [0.05, 0.3] |
| Masking probability | 0.3 |
| Rotation max angle | $180°$ |
| Unit cell perturbation $\sigma$ | [0.01, 0.03] |
| *Random graph augmentation* | |
| Radius range | [4.0, 6.0] Å |
| Max neighbors range | [20, 120] |
| *Compositional stream* | |
| Transformer layers | 4 |
| Number of Heads | 8 |
| Hidden dim | 1024 |
| $d_{\text{comp}}$ | 256 |
| *Structural stream* | |
| GNN layers | 3 |
| Hidden dim | 1024 |
| RBF features | 8 |
| $d_{\text{struct}}$ | 256 |

*Table 20.* Fine-tuning hyperparameters for experiments in Table 1 and Figure 3. These were selected to optimize energy and force performance on a baseline eSEN-conservative model without pretraining.

| Hyperparameter | Value |
|---|---|
| Dataset | 4M samples from OMat24 |
| Steps | 100,000 |
| Batch size | 72 |
| Optimizer | AdamW |
| Learning rate | $3 \times 10^{-4}$ |
| LR schedule | Linear warmup + cosine decay |
| Warmup steps | 500 |
| Weight decay | $1 \times 10^{-3}$ |
| Gradient clipping | 1.0 |
| $\lambda_E$ (energy) | 50 |
| $\lambda_F$ (forces) | 100 |
| $\lambda_\sigma$ (stress) | 50 |
| Precision | float32 |

*Table 21.* Fine-tuning hyperparameters for Matbench Discovery in Table 3.

| Hyperparameter | OMat24 training phase 1 | MPtrj + sAlex phase 2 |
|---|---|---|
| Dataset | OMat24 ( 100M) | sAlex + 8 $\times$ MPtrj |
| Steps | 400,000 | 400,000 |
| Batch size (per GPU) | 128 | 64 |
| Effective batch size (4 GPUs) | 512 | 256 |
| Optimizer | AdamW | AdamW |
| Learning rate | $3 \times 10^{-4}$ | $3 \times 10^{-4}$ |
| LR schedule | Linear warmup + cosine decay | Linear warmup + cosine decay |
| Warmup steps | 500 | 500 |
| Weight decay | $1 \times 10^{-3}$ | $1 \times 10^{-3}$ |
| Gradient clipping | 1.0 | 1.0 |
| $\lambda_E$ (energy) | 50 | 100 |
| $\lambda_F$ (forces) | 100 | 100 |
| $\lambda_\sigma$ (stress) | 50 | 50 |
| Precision | float32 | float32 |

