# OpenReview forum: "TriForces: Augmenting Atomistic GNNs for Transferable Representations"
_ICML.cc/2026/Conference — ICML 2026 regular_

### Official Review · Reviewer_A7R9 · 2026-03-04

**Soundness:** 3
**Presentation:** 4
**Significance:** 3
**Originality:** 4
**Overall Recommendation:** 5
**Confidence:** 3

**Summary:**

The paper proposes a new framework for self-supervised pretraining of MLIPs called TriForces. The core idea is to augment base GNN models such as MACE with composition and structure streams, forming a trio of representations that summarize different parts of the system. The triple stream architecture is additionally pretrained via non-reconstruction, denoising and masking losses, avoiding the use of any DFT labels, improving predictions and latent space construction. The authors demonstrate the capability of TriForces augmented architectures on several transfer learning bechmarks and downstream tasks such as similar structure retrieval.

**Compliance With Llm Reviewing Policy:**

Affirmed.

**Final Justification:**

Thank you to the authors for the rebuttal, and to the AC, SAC, and PCs for handling the discussion.

Based on the paper and rebuttal, my current assessment is as follows:

The paper is well written, technically solid, and includes a strong range of experiments and supporting analyses.

The authors addressed my concerns in the rebuttal well, clearing up my misunderstandings around the backbone-only pretraining and the use of attention networks. And by providing the SOAP baseline in the rebuttal, they better demonstrate TriForces capacity. Hence, while the rebuttal provides general clarity, I still believe that the retrieval usefulness is somewhat limited and that the main contribution are the improved MLIPs. Therefore, I will retain my previous score.

Overall, I think the impact and originality of the paper is good, with some limitations in the retrieval task and data scaling. Overall, it constitutes a good addition to the field and provides a drop-in improvement for many, widely used papers.

**Key Questions For Authors:**

- In the retrieval benchmark, how does TriForce compare to SOAP or another simple kernel baseline?
- Do you expect this pretraining scheme to work for attention based architectures as well?
- Is it possible to directly pretrain one of the baseline models with the TriForces labels? Did you attempt to do so?

**Limitations:**

Yes

**Strengths And Weaknesses:**

## Strengths: ##
The paper is very well written. Each section and subsection clearly describe what they need to without additional fluff. The ideas presented are easy to follow and allow for a good understanding even on the first pass. The figures are appropriate and provide a clean intuition of the architecture and plots and tables are easy to read.

The proposed TriForce architecture is a novel way to promote additional learning without attempting to overload already highly optimized message passing schemes. And the pretraining setup is an excellent way of exploiting working methods without relying on any additional DFT information.

The experiments and ablation studies conducted cover a good variety of possible setups and underscore TriForces benefits.

## Weaknesses: ##
The paper demonstrates that pretraining TriForce augmented GNNs with the three SSL targets is very beneficial,  but does not thoroughly discuss pretraining the baselines themselves. Can the baseline models also be pretrained like TriForce and what outcome do you expect?

Similarly, the retrieval evaluation not clear. How does TriForce compare to SOAP or another simple kernel baseline?

Lastly, while the chosen models are largely representative, a comparison on a transformer or transformer-style GNN is missing. Do you expect similar gains from the composition stream when the model also uses attention for message passing (like EscAIP or Equiformer2)?

---

> ### Author Rebuttal · Authors · 2026-03-30
>
> We thank the reviewer for the clear and actionable suggestions, particularly on retrieval baselines and backbone comparisons.
>
> ---
>
> ### Q1: Comparison to SOAP retrieval baseline
>
> > *In the retrieval benchmark, how does TriForces compare to SOAP or another simple kernel baseline?*
>
> We evaluated two standard handcrafted baselines, `SOAP` and `SineMatrix`, on the same retrieval setup. For TriForces, we follow the paper protocol and use the composition stream for element-set retrieval and the structure stream for space-group retrieval. We use global descriptors (DScribe, periodic), averaging per-site SOAP features.
>
>  Method | Element Set R@10 | Space Group R@10 |
> | --- | ---: | ---: |
> | SOAP (mu1nu1) | **0.814** | 0.103 |
> | SineMatrix (eigenspectrum) | 0.041 | 0.221 |
> | eSEN 30M OAM | 0.184 | 0.207 |
> | ORB-v3 OMat | 0.179 | 0.413 |
> | TriForces eSEN | 0.770 | **0.648** |
> | TriForces ORB-v3 | 0.774 | 0.642 |
>
> SOAP is very strong for element-set retrieval but fails on structure, while SineMatrix shows the opposite trend. TriForces matches SOAP on chemistry and substantially improves structure retrieval, performing strongly on both tasks. This supports the benefit of separating composition and geometry.
>
> ---
>
> ### Q2: Attention based architectures with this framework
>
> > *Do you expect this pretraining scheme to work for attention based architectures as well?*
>
> Yes. The framework is orthogonal to the backbone. For example, ORB-v3 already uses attention in its Message Passing Neural Networks, and we observe consistent gains when augmenting it with TriForces. The composition stream provides a global summary over elements and counts, while the structure stream encodes invariant geometry; both are complementary to atom-level attention. We expect similar behavior for graph-transformer models like EquiformerV2, which we leave for future work due to training cost.
>
> ---
>
> ### Q3: Pretrain backbone-only models with SSL protocol
>
> > *Is it possible to directly pretrain one of the baseline models with the TriForces labels? Did you attempt to do so?*
>
> Yes. We pretrain backbone-only models with the same SSL objectives (Fig. 4, Table 5), isolating SSL from the added streams. SSL alone improves representations (e.g., +15% crystal-system accuracy and ~35% lower energy per atom MAE), but gains are consistently larger with the full three-stream architecture, particularly for geometry-sensitive probes and fine-tuning (Table 5). We clarify this comparison in the main text.

---

> > ### Author Rebuttal · Reviewer_A7R9 · 2026-04-01
> >
> > Thank you for your response. The questions were addressed well and I feel confident in my score and keep it as such.

---

### Official Review · Reviewer_M8MS · 2026-03-10

**Soundness:** 3
**Presentation:** 2
**Significance:** 3
**Originality:** 3
**Overall Recommendation:** 4
**Confidence:** 4

**Summary:**

This paper introduces TriForces, a model-agnostic three-stream framework designed to improve the transferability and reusability of representations in atomistic Graph Neural Networks (GNNs). By explicitly separating composition, structure, and interaction information into parallel streams, and applying a multi-objective self-supervised learning (SSL) pretraining strategy (denoising, masking, and LeJEPA), the authors tackle the common issue of transferability in standard Machine Learning Interatomic Potentials (MLIPs) with fine-tuning. The proposed method is evaluated across multiple backbone architectures (eSEN, Orb-v3, MACE) on benchmarks such as OMat24, MatBench, and QM9, demonstrating improved data efficiency, robust transfer performance, and enabling interpretable similarity retrieval.

**Compliance With Llm Reviewing Policy:**

Affirmed.

**Final Justification:**

This paper provides a valuable systematic analysis of SSL pretraining within the MLIP framework, offering practical guidance on transferability, a timely and underexplored topic. The experimental scope and clarity are strengths.

My main concern, precisely disentangling why SSL outperforms supervised pretraining as initialization, was partially but not fully addressed by the rebuttal. The additional LeMat-Bulk experiment is appreciated, but the evidence remains limited to a single backbone/dataset, and some comparisons (e.g., UMA baselines) are not strictly controlled, as the authors acknowledge.

Despite some concerns regarding the broader necessity of this research direction, I believe this work will inspire many researchers in the community and serve as a strong foundation for future follow-up studies.

**Key Questions For Authors:**

**Questions**

- **Justification of SSL vs. Supervised Scaling with Abundant DFT Labels:**
    - I conceptually agree that decoupling architectural streams combined with SSL is a promising and necessary direction toward building true representation-level foundational models for atomistic systems. However, I would like to hear the authors' perspective on the practical motivation for prioritizing SSL in the current data landscape, which appears somewhat artificial:
        - The "unlabeled" structures used for SSL (e.g., LeMat-Bulk) are typically DFT-relaxed equilibrium structures or sampled from DFT-MD trajectories. Since obtaining these precise geometries inherently requires expensive DFT calculations, the energy and force labels are naturally computed and already available alongside the positions.
        - **Question 1-1 (Practicality):** Given that these labels are already "paid for" during the generation of the structural data, what is the specific practical advantage of discarding them to perform SSL? Is there a scenario in the near future where we would have access to high-quality atomic structures without their corresponding DFT labels?
        - **Question 1-2 (Synergy vs. Trade-off):** When the model is trained using the same dataset for "Supervised Pre-training" (utilizing the available DFT labels) instead of only SSL, does the performance improve or actually degrade? If there is a performance gain, what is the specific rationale for intentionally excluding those labels during the pre-training phase?
        - **Question 1-3 (Scaling vs. SSL):** If a foundational model were scaled up using robust multi-task supervised mixing—utilizing all readily available energy/force labels—does the severe transferability issue still persist to a degree that necessitates a separate SSL phase?
- **Extensibility to Generative Models:** How applicable is the TriForces representation framework to generative models (e.g., Flow Matching or Diffusion models like REPA/RAE)? Do you think is it beneficial to the generative model?
- **Probing Ablation**: In the probing experiments (Figure 8), what is the performance of the backbone model when probed *without* the structure embedding model? Clarifying this would help isolate the exact contribution of the SSL vs. the architectural streams.

**Limitations:**

yes

**Strengths And Weaknesses:**

**Strengths**

- **Model-Agnostic Verification:** The plug-and-play nature of the three-stream architecture is highly commendable. Demonstrating consistent improvements across diverse backbones (both equivariant like MACE/eSEN and non-equivariant like Orb-v3) strongly validates the framework's broad applicability.
- **Mitigation of Energy-Force Loss Coupling:** The paper provides a compelling solution (and theoretical intuition in Appendix A.4) to the difficult problem of balancing energy and force loss weights during conservative training. TriForces reduces this dependency, leading to faster energy convergence without sacrificing force accuracy.
- **Thorough Ablation on Node Features:** The detailed ablation studies demonstrating the relative sensitivity and contribution of each stream (composition vs. structure) opens up promising avenues for future research to build upon these decoupled representations.
- **Interpretable Retrieval:** Disentangling the latent space to allow targeted searches (e.g., retrieving by purely chemical or purely structural similarity) in MLIP framework is a novel contribution to materials discovery workflows.


**Weaknesses / concerns**

- **Fairness of the Baseline Comparison:**
    - **Data exposure / Training strategy (Table 1, 4)**: The comparison in Table 1, 4 raises a natural concern regarding data exposure. The baseline models are trained from scratch on the downstream dataset, whereas the TriForces variants are pre-trained on the LeMat-Bulk dataset as I understand. Since TriForces has seen a significantly larger amount of data, a performance gap is naturally expected. To isolate the benefits of the proposed architecture and SSL from the simple advantage of additional data exposure, it would be necessary to compare against a baseline that is also pre-trained on LeMat-Bulk (e.g., via supervised learning). Furthermore, the original training recipe for foundational models like Orb-v3 involves generative pre-training (e.g., denoising diffusion) prior to fine-tuning. It is unclear why the baselines in Table 1 were not trained following their established pre-training strategies. Evaluating the baselines using their native pre-training recipes would provide a more rigorous and fair comparison.
    - **Parameter Budget and Capacity Trade-offs (Table 1, 3, 4):** TriForces inherently introduces more parameters via the additional streams. While Appendix B.6 provides a parameter-matched baseline showing that TriForces still outperforms standard models, this is a critical aspect that should be elevated to the main text. It must be clear in the main benchmark evaluations (e.g., alongside Table 2 and Table 3) that the performance gains stem fundamentally from the three-stream inductive bias and SSL objectives, not merely from the increased parameter capacity. While I trust that the gains are not solely a function of increased capacity, adding these parameter-matched (or comparable) comparisons to the main tables would significantly strengthen the claims.
- **In-Domain Performance and Practicality at the Foundational Scale:**
The evaluation focuses almost exclusively on transfer performance and low-data regimes. However, the paper leaves several critical questions unanswered regarding its applicability and practicality in large-scale, foundational settings:
    - **In-Domain Trade-offs:** There is a lack of results showing how TriForces performs when trained with *both* SSL and full, large-scale DFT labels on a primary domain (e.g., the entirety of OMat24). Does enforcing these auxiliary SSL objectives and stream separations degrade, or at least not harm, the foundational in-domain predictive accuracy of energy and force? It is crucial to clarify if there is a trade-off between gaining retrieval/probing capabilities and maintaining peak performance.
    - **Transferability at Scale:** The current baselines for transferability are models trained from scratch or on limited data. How does the *energy and force* transferability of TriForces compare to standard MLIPs when fully supervised pre-trained on massive datasets (e.g., tens of millions of structures)? It would significantly strengthen the paper to demonstrate that this structural decoupling provides a transferability edge even when vast amounts of supervised data are available.
    - **Applicability to Already-Trained Models:** The proposed pipeline requires pre-training the entire architecture from scratch using SSL before task-specific fine-tuning. In practice, requiring researchers to re-train foundational models from scratch just to gain transferable representations is somewhat restrictive. Could the authors discuss whether this three-stream augmentation can be retrofitted or fine-tuned onto *already-trained* foundational MLIPs (e.g., freezing or adapting existing interaction weights) without necessitating a full SSL pre-training from scratch?
- **Clarity of Experiment Presentation and Terminology:**
While the extensive evaluation is highly appreciated, tracking the exact training pipelines and terminology across different baselines and TriForces variants is challenging. Improving the presentation of these details will prevent reader confusion and make it a much stronger paper.
    - **Ambiguous Terminology & Settings:** The exact differences between terms like "TriForces", "TriForces-Stream", and "SSL" are sometimes left for the reader to deduce. Furthermore, it is not always explicitly clear in the text and main tables whether a specific baseline is trained from scratch or pre-trained. Explicitly defining these variants early on and consistently labeling the baseline states (e.g., random init vs. pre-trained) across all tables is highly recommended.
    - **Pipeline Summary:** Please add a comprehensive summary table in the appendix detailing the exact pipeline (pre-training data/steps $\rightarrow$ fine-tuning data/steps) for all compared models.
    - **Algorithmic Clarity:** Including an Algorithm block (pseudocode) for both the SSL pre-training and fine-tuning phases in the appendix is highly recommended to facilitate a clearer understanding of the methodology.
    - **Baselines:** Please ensure all benchmark models and datasets are thoroughly cited and detailed in the appendix.

---

++ Having written the review, I realized that the rebuttal period is not very long. I think the necessity of the research would come across more convincingly if the authors could supplement it with experimental results they already have or additional experiments feasible within their available resources. I'm not asking for overly extensive experiments—rather, I'd like to see a clearer explanation of why SSL becomes essential even when we have quality labeled data and pre-trained foundation model, along with a comparison against direct supervised training.

---

> ### Author Rebuttal · Authors · 2026-03-30
>
> We thank the reviewer for the thoughtful and detailed feedback, particularly regarding the role of SSL, fairness of comparisons, and practical considerations.
>
> ---
>
> ### SSL vs supervised pretraining
>
> We agree with the reviewer's comments that SSL is not a replacement for large-scale supervised pretraining. Our claim is narrower: SSL combined with stream separation provides a more transferable initialization, especially when downstream data is limited or when transfer targets differ from the source domain.
>
> #### Q1-1:
>
> > *Scenario in the near future for high-quality atomic structures without DFT labels?*
>
> Yes, for example, generative pipelines or MLIP-relaxed candidate structures can produce large numbers of geometries where DFT labels remain costly. And we observe consistent improvements when applying SSL to non-equilibrium OMat24-1M, even when labels are available. The Table below shows the impact of different pre-training datasets and strategies on down-stream performance.
>
> | TriForces Pre-training | ORB E ↓ | ORB F ↓ | eSEN E ↓ | eSEN F ↓
> | --- | ---: | ---: | ---: | ---: |
> | Supervised LeMat-Bulk | - | - | 22.5 | 106.0 |
> | SSL on LeMat-Bulk | - | - | 18.8 | 86.8 |
> | SSL on 1M OMat | 21.9 | 112.5 | 21.1 | 94.2 |
> | SSL on 10M OMat | 19.8 | 103.6 | 20.4 | 93.8 |
> | SSL on 100M OMat | 18.9 | 99.5 | 18.7 | 87.1 |
>
>
> #### Q1-2:
>
> > *Would supervised pretraining improve performance?*
>
> This is a good point. Table 3 partially addresses this; we also add a controlled comparison showing that on LeMat-Bulk, pretraining (float32 with similar compute budget) advantage shows limited gains for forces likely due to weak force signals in near-equilibrium data (Table above).
>
> > *Why intentionally exclude labels during pretraining?*
>
> On the same setup, SSL yields equal or better downstream performance, especially for forces. Supervised pretraining biases the model toward the regression targets, while SSL encourages learning more general composition and geometric representations. We agree that labels are useful during downstream fine-tuning. Importantly, SSL provides a more transferable initialization.
>
> > *Does SSL harm large-scale supervised accuracy?*
>
> We do not observe degradation: when fine-tuned on full OMat24, TriForces eSEN-sm remains competitive at scale. Experiments on Matbench (Table 2) show performance comparable to the larger and more expressive eSEN model, while being significantly more efficient (>5× faster training and memory footprint).
>
> #### Q1-3:
>
> > *If a foundational model were scaled up with robust multi-task supervision, does the transferability issue persist?*
>
> Recent work (e.g., MACE-MH [1], UMA [2]) shows that scaling supervised multi-task training improves performance. Our results suggest that supervised scaling alone may not fully resolve transferability.
> To test this, we fine-tuned a large supervised UMA-sm model: despite much larger labeled pretraining, transfer remains task-dependent. UMA improves on only 3/10 QM9 targets and 3/8 Matbench tasks relative vs TriForces. While not strictly apples-to-apples (eSEN and UMA are architecturally close but not identical), this suggests that scaling alone may not fully resolve transferability.
>
>
> > *Evaluating models with their native pretraining*
>
> Native pretraining recipes differ across model families in data, objectives, and compute, which would confound comparisons. We therefore use a standardized training pipeline to isolate the contribution of TriForces.
>
> > *Adding parameter-matched to the main tables would significantly strengthen the claims*
>
> We agree this is important and will move the parameter-matched comparisons from Appendix B.6 into the main results tables to clearly show gains are not due to increased capacity.
>
> > *Requiring researchers to re-train foundational models from scratch ... is restrictive*
>
> This is a relevant concern. We study TriForces as a pretraining framework for controlled comparison, but do not see a conceptual obstacle to augmenting existing backbones and continuing with SSL or fine-tuning. We add this discussion.
>
> ---
>
> ### Extensibility to Generative Models
>
> This is a promising direction pointed by the reviewer: separating composition and structure makes TriForces a likely suitable encoder for generative models. We leave this for future work.
>
> ---
>
> ### Probing Ablation
>
> > *Performance of the backbone model when probed without the structure embedding model?*
>
> Fig. 4 isolates SSL (backbone vs TriForces under the same training), while Table 5 shows removing the structure stream degrades geometry-sensitive probing, indicating gains are not due to SSL alone.
>
> We also revise naming conventions for clarity by defining all variants, marking pretraining vs random initialization, and adding a pipeline summary and pseudocode.
>
> [1] Batatia, et al. Cross learning between electronic structure theories for unifying molecular, surface, and inorganic crystal foundation force fields.
>
> [2] Wood, et al. Uma: A family of universal models for atoms.

---

> > ### Author Rebuttal · Reviewer_M8MS · 2026-04-03
> >
> > We thank the authors for the thorough rebuttal and additional experiments.
> >
> > The SSL vs. supervised pretraining comparison on LeMat-Bulk is appreciated. However, I still find it difficult to precisely disentangle why SSL provides a stronger initialization than supervised pretraining — the evidence remains limited to a single backbone and dataset, and the UMA comparison for Q1-3 is not strictly controlled as the authors acknowledge. I believe this distinction deserves further investigation.
> >
> > Nonetheless, I highly value this work's systematic analysis of how to apply SSL within the MLIP framework and its impact on transferability. I maintain my current score.

---

### Official Review · Reviewer_nx7r · 2026-03-12

**Soundness:** 2
**Presentation:** 3
**Significance:** 3
**Originality:** 3
**Overall Recommendation:** 4
**Confidence:** 4

**Summary:**

Authors present a self-supervised learning framework for atomistic GNNs that separates representation into three distinct component models. Compositional information is encoded by a transformer, positional information is encoded by an invariant message passing model, and an equivariant GNN backbone is provided with both element identities and positions. Embeddings these three streams are concatenated and subject to a denoising, masking, and regularization loss (SIGReg). The resulting pretrained models are then fine-tuned on various tasks and compared with well known pretrained equivariant GNN architectures. Authors demonstrate strong results with their Triforce models that have comparatively fewer parameters.

**Compliance With Llm Reviewing Policy:**

Affirmed.

**Final Justification:**

I thank the authors for their constructive rebuttal, which has resolved some of my questions. Overall, the paper presents a an interesting idea with some downstream performance. However, the authors' breakdown confirmed my primary remaining reservation: the empirical gains stem overwhelmingly from the architectural changes rather than the proposed self-supervised learning (SSL) pre-training method itself. Because the core proposed method (SSL) does not drive the majority of the demonstrated improvements, I do not feel a score increase is warranted. Nevertheless, the architectural insights possess clear merit for the community, so my prior assessment is reinforced and I am maintaining my score of a Weak Accept.

**Key Questions For Authors:**

1) Can you answer points a and b?

2) Can you clearly explain the impact of the architecture vs the method? You may have implied this in the work, but it is not clear enough for readers.

3) In figure 4. you show linear probe results for your methods ‘TriForces’ and ‘Orb-v3 Base’, however ‘Orb-v3 Base’ has no associated SSL method? What is this plot depicting? What architecture is being used as ‘TriForces’?

**Limitations:**

Yes

**Strengths And Weaknesses:**

Authors do a good job of introducing prior work, claims are reasonably supported, and results appear to be significant.

However, a few points are unclear, and some terms need to be more explicitly defined:

a) In figure 3 and elsewhere, ‘TriForces-Streams sSEN’ and ‘TriForces eSEN” need to be clearly defined. If this is simply the difference between adding additional architecture to eSEN and pretraining with SIGreg/masking/denoising, it would appear to suggest that most of the benefit is coming from additional architecture and not training method.

b) In equation 9 and 10, what does the first subscript number for $h$ is not described. If these are ‘views,’ please specify in detail how the ‘views’/augmentations are created. If not, please explain this annotation.

Additionally, it is difficult to understand the exact impact of each component in the proposed framework. Authors present a good number of ablations in table 5, but the presentation makes these results very difficult to interpret for two reasons:

c) Most metrics are regression errors, which makes the meaning of a positive percent value somewhat unclear. Readers should not have to guess that +11% means that the error has actually decreased 11%. This is even more confusing with one of the four evaluation metrics being a classification task (+11% here means something entirely different).

d) There is not really a clear trend in overall improvement as components are removed. I would have preferred to see a sequential addition or removal of components.

As currently presented, the results appear to suggest that the addition of the ‘streams’ architectural components are the primary cause for downstream improvements. Further, this work is missing a baseline for the concatenated stream model without SSL pre-training.

---

> ### Author Rebuttal · Authors · 2026-03-30
>
> We thank the reviewer for the positive assessment and for the questions on terminology and the role of architecture vs SSL.
>
> ### Q1(a): Definition of variants
>
> Yes, the reviewer’s interpretation is correct, and we agree that the original manuscript did not make this distinction sufficiently explicit.
> For clarity, we define:
> - TriForces-Streams: three-stream architecture trained from random initialization (no SSL),
> - TriForces: same architecture with SSL pretraining.
>
> We will use this terminology more consistently throughout the paper by employing random init. and SSL to clearly show the differences.
>
> > *It would appear to suggest that most of the benefit is coming from additional architecture and not training method*
>
> This is a valid point and directly relates to the distinction between architecture and SSL. In short, the architecture accounts for most of the gains, while SSL provides additional improvements; we detail this in Q2 below.
>
> ---
>
> ### Q1(b): Augmentations / views
>
> > *In equation 9 and 10, what does the first subscript number for $h$ is not described.*
>
> Yes, the subscripts in equation 9 and 10 denote the two augmented views. Specifically, $h_1$ and $h_2$ correspond to representations computed from two stochastic augmentations of the same input structure.
>
> > *Please specify how the views/augmentations are created*
>
> The augmentations are sampled from:
> (i) atom-type masking,
> (ii) small position perturbations,
> (iii) graph perturbations (cutoff / max-neighbor changes), and
> (iv) random rotations for non-equivariant backbones.
>
> We use moderate perturbation intensities to ensure the task remains informative while avoiding trivial identity mappings.
>
> Following the reviewer's remark, we will add a description of the augmentation procedure in the main text near equations 9 and 10. The hyperparameters for all of these augmentations are currently reported in Appendix D.2.
>
> ---
>
> ### Q1(c,d): Ablation clarity
>
> We agree that the reader should not guess the meaning of the percentage sign for error metrics.
> In Table 5, a positive percentage indicates an improvement relative to the base model, which for MAE means a reduction in error. We will report absolute improvements instead of percentages to avoid ambiguity. Full results are also available in Table 11.
>
> To clarify this, we provide a simplified summary of the stream contributions in our response to Reviewer eWtr (Q3), showing absolute values and their effects.
>
> ---
>
> ### Q2: Impact of architecture vs method
>
> > *Can you clearly explain the impact of the architecture vs the method?*
>
> The primary improvement comes from the three-stream architecture; SSL provides additional gains.
>
> Our ablations show:
>
> - Adding the streams without SSL already yields the largest gains (e.g., Table 1: 104 → 20.8 meV/atom on eSEN).
> - SSL is complementary and provides consistent improvements, which increase with pretraining data and are most pronounced in low-data transfer (Figure 3) and probing (Table 5).
>
> > *This work is missing a baseline for the concatenated stream model without SSL pre-training*
>
> The concatenated stream model without SSL (random init.) corresponds to the “TriForces-Streams” (or “+ Streams”) variant reported in Table 1 and subsequent experiments.
>
> ---
>
> ### Q3: Clarifying Linear probe plot
>
> Figure 4 isolates the effect of the three-stream architecture under identical SSL training. Both models use the same SSL pretraining procedure: “Orb-v3 Base” is the backbone alone, while “TriForces” is the same backbone augmented with the two additional streams. We observe that SSL-pretrained representations are significantly more informative when combined with the three-stream architecture than with the backbone alone, particularly for probing non-supervised features. We will make this distinction explicit in the caption and main text.

---

> > ### Author Rebuttal · Reviewer_nx7r · 2026-04-07
> >
> > Thank you for the detailed rebuttal. I am satisfied with the proposed manuscript updates regarding the variant definitions, augmentations, and ablations. However, the authors' breakdown confirms that the performance improvements stem primarily from the architecture, not the proposed SSL method. While this work has merit, the empirical gains from the SSL method remain small. Because the core method's results are not strong enough to alter my evaluation, I will maintain my current score.

---

> > > ### Author Response · Authors · 2026-04-07
> > >
> > > We appreciate the reviewer's feedback and the clear assessment of our clarifications.
> > >
> > > We agree that the three-stream architecture, which constitutes a core contribution of the paper, accounts for the majority of gains in large-scale supervised settings. At the same time, the paper focuses on transferability, data efficiency, and representation quality, where SSL plays a more significant role, particularly in low-data and transfer settings.
> > >
> > > - In low-data regimes, the gap between TriForces with and without SSL is largest (Fig. 3 and Table 8), with ~19% and ~16% improvements at 20K samples for energy and forces, respectively.
> > > - SSL improves representation quality, as shown by probing and latent space organization (Fig. 4), which are not captured by standard supervised metrics.
> > > - Capabilities such as similarity retrieval (Section 4.5) depend on these structured representations and degrade without SSL, even when the architecture or supervised pretraining is present (Table 5 and Table in Q1-1 of our response to Reviewer M8MS).
> > >
> > > As noted by the reviewer, the marginal benefit of SSL decreases in the large-data regime, where supervised signals dominate. We therefore view the contributions as distinct and regime-dependent: the architecture provides the primary inductive bias, while SSL is most beneficial for transfer, representation structure, and downstream analysis.
> > >
> > > Further disentangling the mechanisms by which SSL improves initialization compared to supervised pretraining remains an important direction for future work, and we will clarify the empirical evidence in the revision.
> > >
> > > We thank the reviewers for their constructive feedback and the opportunity to clarify our positioning: TriForces combines a three-stream architecture with multi-objective SSL, where the architecture drives large-data performance and SSL improves transfer, data efficiency, and representation quality.

---

### Official Review · Reviewer_eWtr · 2026-03-12

**Soundness:** 4
**Presentation:** 4
**Significance:** 3
**Originality:** 3
**Overall Recommendation:** 4
**Confidence:** 4

**Summary:**

This paper proposes TriForces, a model-agnostic three-stream (composition, structural, and interaction streams) architecture for MLIPs, combined with SSL pretraining (LeJEPA, Denoising, Masking), which significantly improves MLIP performance on transfer tasks.

**Compliance With Llm Reviewing Policy:**

Affirmed.

**Final Justification:**

i will keep my positive score

**Key Questions For Authors:**

1.	In Figure 2, the gains of TriForces-Streams and TriForces decrease as the dataset size grows. I am concerned whether these methods can still provide benefits in the large-data regime.

2.	Figure 2 also shows that the performance of TriForces-Streams and TriForces becomes increasingly similar with larger datasets. Does this mean that the SSL pretraining may no longer be necessary? As noted in [1], using SSL (DeNS) on OMat24-100M did not improve performance. I am curious whether TriForces’ SSL would reach the same conclusion. This is important for the development of pretrained MLIPs, as it could save the cost of SSL training.

3.	See Weakness 3. Could the authors train each stream individually and compare it to the full TriForces (without SSL) on the corresponding tasks? Specifically, I would like to know whether any two streams benefit the third stream.

[1] Barroso-Luque et al. Open Materials 2024 (OMat24) Inorganic Materials Dataset and Models

**Limitations:**

yes

**Strengths And Weaknesses:**

Strengths:

1.	The paper is well-structured, clearly written, and the figures and tables are clear and well-designed.

2.	It is soundness that TriForces separates graph information into composition, structural, and interaction streams and learns them independently.

3.	The experiments on downstream applications are thorough, covering multiple datasets and multiple models.

Weaknesses:

1.	I think the main contribution of the paper is the model-agnostic three-stream framework; the SSL pretraining is an application of existing techniques.

2.	Figure 2 shows that as the dataset size increases, the gains of TriForces decrease, especially for forces. It would be valuable to conduct ablation studies on TriForces in the large-data regime (e.g., using OMat24-100M) to better understand its behavior.

3.	In Figure 5, I acknowledge that the composition and structural streams support and enhance the interaction stream. However, it is unclear whether any two streams enhance the third stream.

---

> ### Author Rebuttal · Authors · 2026-03-31
>
> We thank the reviewer for the positive assessment, particularly regarding clarity and the value of the three-stream factorization. We also appreciate the questions on scaling behavior and the role of SSL in the large-data regime.
>
> ---
>
> ### Q1: Behaviour in the large-data regime
>
> > *The gains of TriForces decrease as the dataset size grows… can these methods still provide benefits in the large-data regime?*
>
> We agree that gains shrink as data increases. This is expected: TriForces is primarily designed to improve transferability and data efficiency, where standard MLIPs struggle most, which constitutes the main contribution of this work.
>
> Moreover, our results indicate that the three-stream architecture remains beneficial at scale, although gains are smaller than in low-data regimes. Notably, in the Matbench Discovery experiments (Table 3), TriForces achieves lower relaxation energy MAE and comparable RMSD after large-scale training (full OMat24, followed by sAlex and MPtrj). Importantly, these gains are obtained with a model that remains substantially smaller than the large eSEN baseline (under 12M parameters for TriForces, versus 30M for eSEN, while requiring over 5x less training time and memory).
>
> Importantly, we also observe (response to Rev. M8MS, Table in Q1-1) that increasing the amount of pretraining data (from 1M to 100M structures) continues to improve downstream performance, indicating that representation quality benefits from scale even beyond the supervised regime.
>
> ---
>
> ### Q2
>
> > *Does this mean that the SSL pretraining may no longer be necessary at scale?*
>
> At large supervised data scale, SSL is no longer the main source of improvement: most of the gain comes from the three-stream architecture, while SSL provides only modest additional benefits. This is consistent with the reviewer’s observation that SSL gains shrink at scale. Importantly, SSL provides its largest benefits in the low-data and transfer regimes that motivate this work (Table 1 and 3), where it improves both accuracy and representation quality (Figure 4). It also enables tasks like similarity retrieval (Figure 5), which perform poorly with purely supervised scaling.
> To assess this, we ran an additional preliminary comparison in the large-data regime (full OMat24, >110M structures) between (i) TriForces trained from random initialization (TriForces-Streams) and (ii) TriForces with SSL pretraining.
>
> At this scale, SSL yields only marginal improvements in final accuracy (improves by 0.2 meV/atom and 0.1 meV/Å in our preliminary run), but still improves convergence speed.
>
> | Setting | E/atom (meV/atom) ↓ | F (meV/Å) ↓ |
> | --- | ---: | ---: |
> | TriForces eSEN (random init) | 13.7 | 62.2 |
> | TriForces eSEN (SSL) | 13.5 | 62.1 |
>
> This indicates that SSL is not essential at very large supervised scale; its main value lies in transfer and lower-data regimes.
>
>
> ---
>
> ### Q3: Stream interactions
>
> > *Whether any two streams enhance the third stream*
>
> The streams are complementary rather than uniformly redundant: composition primarily improves energy prediction, interaction is critical for forces and stress, and structure contributes most to geometric probing and representation quality (Table 5 and Figure 5). Following the reviewer’s suggestion, we ran an ablation on OMat24-2M without SSL using an enegy-conserving eSEN-sm backbone. Rather than uniform gains, each stream contributes to different aspects, and the full model provides the best overall balance.
>
> | Transition | E/atom MAE ↓ (meV/atom) | F MAE ↓ (meV/Å) | S MAE ↓ (meV/Å³) | Probe Crystal System Acc ↑ | Probe E/atom MLP MAE ↓ (meV/atom) |
> |---|---:|---:|---:|---:|---:|
> | Interaction → Composition + Interaction | 102 → 16 | 79 → 77 | 6.1 → 4.3 | 44.5% → 50.6% | 61.4 → 55.0 |
> | Composition + Structure → + Interaction | 68 → 16 | 249 → 78 | 8.9 → 4.4 | 66.1% → 68.1% | 126.6 → 55.4 |
> | Composition + Interaction → + Structure | 16 → 16 | 77 → 76 | 4.3 → 4.3 | 50.6% → 68.1% | 55.0 → 55.4 |
>
> We will revise the text to make these roles explicit and include this ablation in the appendix.

---

> > ### Author Rebuttal · Reviewer_eWtr · 2026-04-03
> >
> > I will keep my positive score.

---

### Decision · Program_Chairs · 2026-04-30

**Decision:**

Accept (regular)

**Comment:**

All reviewers are supportive to this work, and most of issues have been resolved during rebuttals. Thus an accept is recommended.